# SPECTRAL NORMALIZATION
# FOR GENERATIVE ADVERSARIAL NETWORKS

**Takeru Miyato[1], Toshiki Kataoka[1], Masanori Koyama[2], Yuichi Yoshida[3]**
{miyato, kataoka}@preferred.jp
koyama.masanori@gmail.com
yyoshida@nii.ac.jp
[1]Preferred Networks, Inc. [2]Ritsumeikan University [3]National Institute of Informatics

## ABSTRACT

One of the challenges in the study of generative adversarial networks is the instability of its training. In this paper, we propose a novel weight normalization technique called spectral normalization to stabilize the training of the discriminator. Our new normalization technique is computationally light and easy to incorporate into existing implementations. We tested the efficacy of spectral normalization on CIFAR10, STL-10, and ILSVRC2012 dataset, and we experimentally confirmed that spectrally normalized GANs (SN-GANs) is capable of generating images of better or equal quality relative to the previous training stabilization techniques. The code with Chainer (Tokui et al., 2015), generated images and pretrained models are available at https://github.com/pfnet-research/sngan_projection.

## 1 INTRODUCTION

*Generative adversarial networks* (GANs) (Goodfellow et al., 2014) have been enjoying considerable success as a framework of generative models in recent years, and it has been applied to numerous types of tasks and datasets (Radford et al., 2016; Salimans et al., 2016; Ho & Ermon, 2016; Li et al., 2017). In a nutshell, GANs are a framework to produce a model distribution that mimics a given target distribution, and it consists of a generator that produces the model distribution and a discriminator that distinguishes the model distribution from the target. The concept is to consecutively train the model distribution and the discriminator in turn, with the goal of reducing the difference between the model distribution and the target distribution measured by the best discriminator possible at each step of the training. GANs have been drawing attention in the machine learning community not only for its ability to learn highly structured probability distribution but also for its theoretically interesting aspects. For example, (Nowozin et al., 2016; Uehara et al., 2016; Mohamed & Lakshminarayanan, 2017) revealed that the training of the discriminator amounts to the training of a good estimator for the density ratio between the model distribution and the target. This is a perspective that opens the door to the methods of *implicit models* (Mohamed & Lakshminarayanan, 2017; Tran et al., 2017) that can be used to carry out variational optimization without the direct knowledge of the density function.

A persisting challenge in the training of GANs is the performance control of the discriminator. In high dimensional spaces, the density ratio estimation by the discriminator is often inaccurate and unstable during the training, and generator networks fail to learn the multimodal structure of the target distribution. Even worse, when the support of the model distribution and the support of the target distribution are disjoint, there exists a discriminator that can perfectly distinguish the model distribution from the target (Arjovsky & Bottou, 2017). Once such discriminator is produced in this situation, the training of the generator comes to complete stop, because the derivative of the so-produced discriminator with respect to the input turns out to be 0. This motivates us to introduce some form of restriction to the choice of the discriminator.

In this paper, we propose a novel weight normalization method called *spectral normalization* that can stabilize the training of discriminator networks. Our normalization enjoys following favorable properties.

- Lipschitz constant is the only hyper-parameter to be tuned, and the algorithm does not require intensive tuning of the only hyper-parameter for satisfactory performance.
- Implementation is simple and the additional computational cost is small.

In fact, our normalization method also functioned well even without tuning Lipschitz constant, which is the only hyper parameter. In this study, we provide explanations of the effectiveness of spectral normalization for GANs against other regularization techniques, such as weight normalization (Salimans & Kingma, 2016), weight clipping (Arjovsky et al., 2017), and gradient penalty (Gulrajani et al., 2017). We also show that, in the absence of complimentary regularization techniques (e.g., batch normalization, weight decay and feature matching on the discriminator), spectral normalization can improve the sheer quality of the generated images better than weight normalization and gradient penalty.

## 2  METHOD

In this section, we will lay the theoretical groundwork for our proposed method. Let us consider a simple discriminator made of a neural network of the following form, with the input $\boldsymbol{x}$:

$$f(\boldsymbol{x}, \theta) = W^{L+1} a_L(W^L(a_{L-1}(W^{L-1}(\dots a_1(W^1 \boldsymbol{x}) \dots)))), \tag{1}$$

where $\theta := \{W^1, \dots, W^L, W^{L+1}\}$ is the learning parameters set, $W^l \in \mathbb{R}^{d_l \times d_{l-1}}$, $W^{L+1} \in \mathbb{R}^{1 \times d_L}$, and $a_l$ is an element-wise non-linear activation function. We omit the bias term of each layer for simplicity. The final output of the discriminator is given by

$$D(\boldsymbol{x}, \theta) = \mathcal{A}(f(\boldsymbol{x}, \theta)), \tag{2}$$

where $\mathcal{A}$ is an activation function corresponding to the divergence of distance measure of the user's choice. The standard formulation of GANs is given by

$$\min_G \max_D V(G, D)$$

where min and max of $G$ and $D$ are taken over the set of generator and discriminator functions, respectively. The conventional form of $V(G, D)$ (Goodfellow et al., 2014) is given by $\mathrm{E}_{\boldsymbol{x} \sim q_{\mathrm{data}}}[\log D(\boldsymbol{x})] + \mathrm{E}_{\boldsymbol{x}' \sim p_G}[\log(1 - D(\boldsymbol{x}'))]$, where $q_{\mathrm{data}}$ is the data distribution and $p_G$ is the (model) generator distribution to be learned through the adversarial min-max optimization. The activation function $\mathcal{A}$ that is used in the $D$ of this expression is some continuous function with range $[0, 1]$ (e.g, sigmoid function). It is known that, for a fixed generator $G$, the optimal discriminator for this form of $V(G, D)$ is given by $D_G^*(\boldsymbol{x}) := q_{\mathrm{data}}(\boldsymbol{x})/(q_{\mathrm{data}}(\boldsymbol{x}) + p_G(\boldsymbol{x}))$.

The machine learning community has been pointing out recently that the function space from which the discriminators are selected crucially affects the performance of GANs. A number of works (Uehara et al., 2016; Qi, 2017; Gulrajani et al., 2017) advocate the importance of Lipschitz continuity in assuring the boundedness of statistics. For example, the optimal discriminator of GANs on the above standard formulation takes the form

$$D_G^*(\boldsymbol{x}) = \frac{q_{\mathrm{data}}(\boldsymbol{x})}{q_{\mathrm{data}}(\boldsymbol{x}) + p_G(\boldsymbol{x})} = \mathrm{sigmoid}(f^*(\boldsymbol{x})), \text{where } f^*(\boldsymbol{x}) = \log q_{\mathrm{data}}(\boldsymbol{x}) - \log p_G(\boldsymbol{x}), \tag{3}$$

and its derivative

$$\nabla_{\boldsymbol{x}} f^*(x) = \frac{1}{q_{\mathrm{data}}(\boldsymbol{x})} \nabla_{\boldsymbol{x}} q_{\mathrm{data}}(\boldsymbol{x}) - \frac{1}{p_G(\boldsymbol{x})} \nabla_{\boldsymbol{x}} p_G(\boldsymbol{x}) \tag{4}$$

can be unbounded or even incomputable. This prompts us to introduce some regularity condition to the derivative of $f(\boldsymbol{x})$.

A particularly successful works in this array are (Qi, 2017; Arjovsky et al., 2017; Gulrajani et al., 2017), which proposed methods to control the Lipschitz constant of the discriminator by adding regularization terms defined on input examples $\boldsymbol{x}$. We would follow their footsteps and search for the discriminator $D$ from the set of $K$-Lipschitz continuous functions, that is,

$$\underset{\|f\|_{\mathrm{Lip}} \leq K}{\arg \max} V(G, D), \tag{5}$$

where we mean by $\|f\|_{\mathrm{Lip}}$ the smallest value $M$ such that $\|f(\boldsymbol{x}) - f(\boldsymbol{x}')\| / \|\boldsymbol{x} - \boldsymbol{x}'\| \leq M$ for any $\boldsymbol{x}, \boldsymbol{x}'$, with the norm being the $\ell_2$ norm.

While input based regularizations allow for relatively easy formulations based on samples, they also suffer from the fact that, they cannot impose regularization on the space outside of the supports of the generator and data distributions without introducing somewhat heuristic means. A method we would introduce in this paper, called *spectral normalization*, is a method that aims to skirt this issue by *normalizing* the weight matrices using the technique devised by Yoshida & Miyato (2017).

## 2.1 SPECTRAL NORMALIZATION

Our spectral normalization controls the Lipschitz constant of the discriminator function $f$ by literally constraining the spectral norm of each layer $g : \boldsymbol{h}_{in} \mapsto \boldsymbol{h}_{out}$. By definition, Lipschitz norm $\|g\|_{\mathrm{Lip}}$ is equal to $\sup_{\boldsymbol{h}} \sigma(\nabla g(\boldsymbol{h}))$, where $\sigma(A)$ is the spectral norm of the matrix $A$ ($L_2$ matrix norm of $A$)

$$\sigma(A) := \max_{\boldsymbol{h}:\boldsymbol{h}\neq\boldsymbol{0}} \frac{\|A\boldsymbol{h}\|_2}{\|\boldsymbol{h}\|_2} = \max_{\|\boldsymbol{h}\|_2\leq 1} \|A\boldsymbol{h}\|_2, \tag{6}$$

which is equivalent to the largest singular value of $A$. Therefore, for a linear layer $g(\boldsymbol{h}) = W\boldsymbol{h}$, the norm is given by $\|g\|_{\mathrm{Lip}} = \sup_{\boldsymbol{h}} \sigma(\nabla g(\boldsymbol{h})) = \sup_{\boldsymbol{h}} \sigma(W) = \sigma(W)$. If the Lipschitz norm of the activation function $\|a_l\|_{\mathrm{Lip}}$ is equal to 1 [1], we can use the inequality $\|g_1 \circ g_2\|_{\mathrm{Lip}} \leq \|g_1\|_{\mathrm{Lip}} \cdot \|g_2\|_{\mathrm{Lip}}$ to observe the following bound on $\|f\|_{\mathrm{Lip}}$:

$$\|f\|_{\mathrm{Lip}} \leq \|(\boldsymbol{h}_L \mapsto W^{L+1}\boldsymbol{h}_L)\|_{\mathrm{Lip}} \cdot \|a_L\|_{\mathrm{Lip}} \cdot \|(\boldsymbol{h}_{L-1} \mapsto W^L\boldsymbol{h}_{L-1})\|_{\mathrm{Lip}}$$

$$\cdots \|a_1\|_{\mathrm{Lip}} \cdot \|(\boldsymbol{h}_0 \mapsto W^1\boldsymbol{h}_0)\|_{\mathrm{Lip}} = \prod_{l=1}^{L+1} \|(\boldsymbol{h}_{l-1} \mapsto W^l\boldsymbol{h}_{l-1})\|_{\mathrm{Lip}} = \prod_{l=1}^{L+1} \sigma(W^l). \tag{7}$$

Our *spectral normalization* normalizes the spectral norm of the weight matrix $W$ so that it satisfies the Lipschitz constraint $\sigma(W) = 1$:

$$\bar{W}_{\mathrm{SN}}(W) := W/\sigma(W). \tag{8}$$

If we normalize each $W^l$ using (8), we can appeal to the inequality (7) and the fact that $\sigma\left(\bar{W}_{\mathrm{SN}}(W)\right) = 1$ to see that $\|f\|_{\mathrm{Lip}}$ is bounded from above by 1.

Here, we would like to emphasize the difference between our spectral normalization and spectral norm "regularization" introduced by Yoshida & Miyato (2017). Unlike our method, spectral norm "regularization" penalizes the spectral norm by adding explicit regularization term to the objective function. Their method is fundamentally different from our method in that they do not make an attempt to 'set' the spectral norm to a designated value. Moreover, when we reorganize the derivative of our normalized cost function and rewrite our objective function (12), we see that our method is augmenting the cost function with a sample data *dependent* regularization function. Spectral norm regularization, on the other hand, imposes sample data *independent* regularization on the cost function, just like L2 regularization and Lasso.

## 2.2 FAST APPROXIMATION OF THE SPECTRAL NORM $\sigma(W)$

As we mentioned above, the spectral norm $\sigma(W)$ that we use to regularize each layer of the discriminator is the largest singular value of $W$. If we naively apply singular value decomposition to compute the $\sigma(W)$ at each round of the algorithm, the algorithm can become computationally heavy. Instead, we can use the power iteration method to estimate $\sigma(W)$ (Golub & Van der Vorst, 2000; Yoshida & Miyato, 2017). With power iteration method, we can estimate the spectral norm with very small additional computational time relative to the full computational cost of the vanilla GANs. Please see Appendix A for the detail method and Algorithm 1 for the summary of the actual spectral normalization algorithm.

---

[1]For examples, ReLU (Jarrett et al., 2009; Nair & Hinton, 2010; Glorot et al., 2011) and leaky ReLU (Maas et al., 2013) satisfies the condition, and many popular activation functions satisfy $K$-Lipschitz constraint for some predefined $K$ as well.

## 2.3 Gradient analysis of the spectrally normalized weights

The gradient[2] of $\bar{W}_{\mathrm{SN}}(W)$ with respect to $W_{ij}$ is:

$$\frac{\partial \bar{W}_{\mathrm{SN}}(W)}{\partial W_{ij}} = \frac{1}{\sigma(W)} E_{ij} - \frac{1}{\sigma(W)^2} \frac{\partial \sigma(W)}{\partial W_{ij}} W = \frac{1}{\sigma(W)} E_{ij} - \frac{[\boldsymbol{u}_1 \boldsymbol{v}_1^{\mathrm{T}}]_{ij}}{\sigma(W)^2} W \tag{9}$$

$$= \frac{1}{\sigma(W)} \left( E_{ij} - [\boldsymbol{u}_1 \boldsymbol{v}_1^{\mathrm{T}}]_{ij} \bar{W}_{\mathrm{SN}} \right), \tag{10}$$

where $E_{ij}$ is the matrix whose $(i, j)$-th entry is 1 and zero everywhere else, and $\boldsymbol{u}_1$ and $\boldsymbol{v}_1$ are respectively the first left and right singular vectors of $W$. If $\boldsymbol{h}$ is the hidden layer in the network to be transformed by $\bar{W}_{SN}$, the derivative of the $V(G, D)$ calculated over the mini-batch with respect to $W$ of the discriminator $D$ is given by:

$$\frac{\partial V(G, D)}{\partial W} = \frac{1}{\sigma(W)} \left( \hat{\mathrm{E}} \left[ \boldsymbol{\delta} \boldsymbol{h}^{\mathrm{T}} \right] - \left( \hat{\mathrm{E}} \left[ \boldsymbol{\delta}^{\mathrm{T}} \bar{W}_{\mathrm{SN}} \boldsymbol{h} \right] \right) \boldsymbol{u}_1 \boldsymbol{v}_1^{\mathrm{T}} \right) \tag{11}$$

$$= \frac{1}{\sigma(W)} \left( \hat{\mathrm{E}} \left[ \boldsymbol{\delta} \boldsymbol{h}^{\mathrm{T}} \right] - \lambda \boldsymbol{u}_1 \boldsymbol{v}_1^{\mathrm{T}} \right) \tag{12}$$

where $\boldsymbol{\delta} := \left( \partial V(G, D)/\partial \left( \bar{W}_{\mathrm{SN}} \boldsymbol{h} \right) \right)^{\mathrm{T}}$, $\lambda := \hat{\mathrm{E}} \left[ \boldsymbol{\delta}^{\mathrm{T}} \left( \bar{W}_{\mathrm{SN}} \boldsymbol{h} \right) \right]$, and $\hat{\mathrm{E}}[\cdot]$ represents empirical expectation over the mini-batch. $\frac{\partial V}{\partial W} = 0$ when $\hat{\mathrm{E}}[\boldsymbol{\delta} \boldsymbol{h}^{\mathrm{T}}] = k \boldsymbol{u}_1 \boldsymbol{v}_1^T$ for some $k \in \mathbb{R}$.

We would like to comment on the implication of (12). The first term $\hat{\mathrm{E}} \left[ \boldsymbol{\delta} \boldsymbol{h}^{\mathrm{T}} \right]$ is the same as the derivative of the weights without normalization. In this light, the second term in the expression can be seen as the regularization term penalizing the first singular components with the *adaptive* regularization coefficient $\lambda$. $\lambda$ is positive when $\boldsymbol{\delta}$ and $\bar{W}_{\mathrm{SN}} \boldsymbol{h}$ are pointing in similar direction, and this prevents the column space of $W$ from concentrating into one particular direction in the course of the training. In other words, spectral normalization prevents the transformation of each layer from becoming to sensitive in one direction. We can also use spectral normalization to devise a new parametrization for the model. Namely, we can split the layer map into two separate trainable components: spectrally normalized map and the spectral norm constant. As it turns out, this parametrization has its merit on its own and promotes the performance of GANs (See Appendix E).

## 3 Spectral Normalization vs Other Regularization Techniques

The weight normalization introduced by Salimans & Kingma (2016) is a method that normalizes the $\ell_2$ norm of each row vector in the weight matrix. Mathematically, this is equivalent to requiring the weight by the weight normalization $\bar{W}_{\mathrm{WN}}$:

$$\sigma_1(\bar{W}_{\mathrm{WN}})^2 + \sigma_2(\bar{W}_{\mathrm{WN}})^2 + \cdots + \sigma_T(\bar{W}_{\mathrm{WN}})^2 = d_o, \text{ where } T = \min(d_i, d_o), \tag{13}$$

where $\sigma_t(A)$ is a $t$-th singular value of matrix $A$. Therefore, up to a scaler, this is same as the Frobenius normalization, which requires the sum of the squared singular values to be 1. These normalizations, however, inadvertently impose much stronger constraint on the matrix than intended. If $\bar{W}_{\mathrm{WN}}$ is the weight normalized matrix of dimension $d_i \times d_o$, the norm $\|\bar{W}_{\mathrm{WN}} \boldsymbol{h}\|_2$ for a fixed unit vector $\boldsymbol{h}$ is maximized at $\|\bar{W}_{\mathrm{WN}} \boldsymbol{h}\|_2 = \sqrt{d_o}$ when $\sigma_1(\bar{W}_{\mathrm{WN}}) = \sqrt{d_o}$ and $\sigma_t(\bar{W}_{\mathrm{WN}}) = 0$ for $t = 2, \ldots, T$, which means that $\bar{W}_{\mathrm{WN}}$ is of rank one. Similar thing can be said to the Frobenius normalization (See the appendix for more details). Using such $\bar{W}_{\mathrm{WN}}$ corresponds to using only one feature to discriminate the model probability distribution from the target. In order to retain as much norm of the input as possible and hence to make the discriminator more sensitive, one would hope to make the norm of $\bar{W}_{\mathrm{WN}} \boldsymbol{h}$ large. For weight normalization, however, this comes at the cost of reducing the rank and hence the number of features to be used for the discriminator. Thus, there is a conflict of interests between weight normalization and our desire to use as many features as possible to distinguish the generator distribution from the target distribution. The former interest often reigns over the other in many cases, inadvertently diminishing the number of features to be used by the discriminators. Consequently, the algorithm would produce a rather arbitrary model distribution

---

[2]Indeed, when the spectrum has multiplicities, we would be looking at subgradients here. However, the probability of this happening is zero (almost surely), so we would continue discussions without giving considerations to such events.

that matches the target distribution only at select few features. Weight clipping (Arjovsky et al., 2017) also suffers from same pitfall.

Our spectral normalization, on the other hand, do not suffer from such a conflict in interest. Note that the Lipschitz constant of a linear operator is determined only by the maximum singular value. In other words, the spectral norm is independent of rank. Thus, unlike the weight normalization, our spectral normalization allows the parameter matrix to use as many features as possible while satisfying local 1-Lipschitz constraint. Our spectral normalization leaves more freedom in choosing the number of singular components (features) to feed to the next layer of the discriminator.

Brock et al. (2016) introduced orthonormal regularization on each weight to stabilize the training of GANs. In their work, Brock et al. (2016) augmented the adversarial objective function by adding the following term:

$$\|W^{\mathrm{T}}W - I\|_F^2. \tag{14}$$

While this seems to serve the same purpose as spectral normalization, orthonormal regularization are mathematically quite different from our spectral normalization because the orthonormal regularization destroys the information about the spectrum by setting all the singular values to one. On the other hand, spectral normalization only scales the spectrum so that the its maximum will be one.

Gulrajani et al. (2017) used Gradient penalty method in combination with WGAN. In their work, they placed $K$-Lipschitz constant on the discriminator by augmenting the objective function with the regularizer that rewards the function for having local 1-Lipschitz constant (i.e. $\|\nabla_{\hat{x}} f\|_2 = 1$) at discrete sets of points of the form $\hat{x} := \epsilon \tilde{x} + (1 - \epsilon)x$ generated by interpolating a sample $\tilde{x}$ from generative distribution and a sample $x$ from the data distribution. While this rather straightforward approach does not suffer from the problems we mentioned above regarding the effective dimension of the feature space, the approach has an obvious weakness of being heavily dependent on the support of the current generative distribution. As a matter of course, the generative distribution and its support gradually changes in the course of the training, and this can destabilize the effect of such regularization. In fact, we empirically observed that a high learning rate can destabilize the performance of WGAN-GP. On the contrary, our spectral normalization regularizes the function the operator space, and the effect of the regularization is more stable with respect to the choice of the batch. Training with our spectral normalization does not easily destabilize with aggressive learning rate. Moreover, WGAN-GP requires more computational cost than our spectral normalization with single-step power iteration, because the computation of $\|\nabla_{\hat{x}} f\|_2$ requires one whole round of forward and backward propagation. In the appendix section, we compare the computational cost of the two methods for the same number of updates.

## 4 EXPERIMENTS

In order to evaluate the efficacy of our approach and investigate the reason behind its efficacy, we conducted a set of extensive experiments of unsupervised image generation on CIFAR-10 (Torralba et al., 2008) and STL-10 (Coates et al., 2011), and compared our method against other normalization techniques. To see how our method fares against large dataset, we also applied our method on ILSVRC2012 dataset (ImageNet) (Russakovsky et al., 2015) as well. This section is structured as follows. First, we will discuss the objective functions we used to train the architecture, and then we will describe the optimization settings we used in the experiments. We will then explain two performance measures on the images to evaluate the images produced by the trained generators. Finally, we will summarize our results on CIFAR-10, STL-10, and ImageNet.

As for the architecture of the discriminator and generator, we used convolutional neural networks. Also, for the evaluation of the spectral norm for the convolutional weight $W \in \mathbb{R}^{d_{\mathrm{out}} \times d_{\mathrm{in}} \times h \times w}$, we treated the operator as a 2-D matrix of dimension $d_{\mathrm{out}} \times (d_{\mathrm{in}}hw)$[3]. We trained the parameters of the generator with batch normalization (Ioffe & Szegedy, 2015). We refer the readers to Table 3 in the appendix section for more details of the architectures.

---

[3]Note that, since we are conducting the convolution discretely, the spectral norm will depend on the size of the stride and padding. However, the answer will only differ by some predefined $K$.

For all methods other than WGAN-GP, we used the following standard objective function for the adversarial loss:

$$V(G, D) := \mathop{\mathrm{E}}_{x \sim q_{\mathrm{data}}(\boldsymbol{x})} [\log D(\boldsymbol{x})] + \mathop{\mathrm{E}}_{\boldsymbol{z} \sim p(\boldsymbol{z})} [\log(1 - D(G(\boldsymbol{z})))], \tag{15}$$

where $\boldsymbol{z} \in \mathbb{R}^{d_z}$ is a latent variable, $p(\boldsymbol{z})$ is the standard normal distribution $\mathcal{N}(0, I)$, and $G : \mathbb{R}^{d_z} \to \mathbb{R}^{d_0}$ is a deterministic generator function. We set $d_z$ to 128 for all of our experiments. For the updates of $G$, we used the alternate cost proposed by Goodfellow et al. (2014) $- \mathrm{E}_{\boldsymbol{z} \sim p(\boldsymbol{z})}[\log(D(G(\boldsymbol{z})))]$ as used in Goodfellow et al. (2014) and Warde-Farley & Bengio (2017). For the updates of $D$, we used the original cost defined in (15). We also tested the performance of the algorithm with the so-called hinge loss, which is given by

$$V_D(\hat{G}, D) = \mathop{\mathrm{E}}_{\boldsymbol{x} \sim q_{\mathrm{data}}(\boldsymbol{x})} [\min(0, -1 + D(\boldsymbol{x}))] + \mathop{\mathrm{E}}_{\boldsymbol{z} \sim p(\boldsymbol{z})} \left[ \min\left(0, -1 - D\left(\hat{G}(\boldsymbol{z})\right)\right) \right] \tag{16}$$

$$V_G(G, \hat{D}) = - \mathop{\mathrm{E}}_{\boldsymbol{z} \sim p(\boldsymbol{z})} \left[ \hat{D}(G(\boldsymbol{z})) \right], \tag{17}$$

respectively for the discriminator and the generator. Optimizing these objectives is equivalent to minimizing the so-called reverse KL divergence : $\mathrm{KL}[p_g || q_{\mathrm{data}}]$. This type of loss has been already proposed and used in Lim & Ye (2017); Tran et al. (2017). The algorithm based on the hinge loss also showed good performance when evaluated with inception score and FID. For Wasserstein GANs with gradient penalty (WGAN-GP) (Gulrajani et al., 2017), we used the following objective function: $V(G, D) := \mathrm{E}_{\boldsymbol{x} \sim q_{\mathrm{data}}}[D(\boldsymbol{x})] - \mathrm{E}_{\boldsymbol{z} \sim p(\boldsymbol{z})}[D(G(\boldsymbol{z}))] - \lambda \, \mathrm{E}_{\hat{\boldsymbol{x}} \sim p_{\hat{\boldsymbol{x}}}}[(\|\nabla_{\hat{\boldsymbol{x}}} D(\hat{\boldsymbol{x}})\|_2 - 1)^2]$, where the regularization term is the one we introduced in the appendix section D.4.

For quantitative assessment of generated examples, we used *inception score* (Salimans et al., 2016) and *Fréchet inception distance* (FID) (Heusel et al., 2017). Please see Appendix B.1 for the details of each score.

## 4.1 RESULTS ON CIFAR10 AND STL-10

In this section, we report the accuracy of the spectral normalization (we use the abbreviation: SN-GAN for the spectrally normalized GANs) during the training, and the dependence of the algorithm's performance on the hyperparmeters of the optimizer. We also compare the performance quality of the algorithm against those of other regularization/normalization techniques for the discriminator networks, including: Weight clipping (Arjovsky et al., 2017), WGAN-GP (Gulrajani et al., 2017), batch-normalization (BN) (Ioffe & Szegedy, 2015), layer normalization (LN) (Ba et al., 2016), weight normalization (WN) (Salimans & Kingma, 2016) and orthonormal regularization (*orthonormal*) (Brock et al., 2016). In order to evaluate the stand-alone efficacy of the gradient penalty, we also applied the gradient penalty term to the standard adversarial loss of GANs (15). We would refer to this method as 'GAN-GP'. For weight clipping, we followed the original work Arjovsky et al. (2017) and set the clipping constant $c$ at 0.01 for the convolutional weight of each layer. For gradient penalty, we set $\lambda$ to 10, as suggested in Gulrajani et al. (2017). For *orthonormal*, we initialized the each weight of $D$ with a randomly selected orthonormal operator and trained GANs with the objective function augmented with the regularization term used in Brock et al. (2016). For all comparative studies throughout, we excluded the multiplier parameter $\boldsymbol{\gamma}$ in the weight normalization method, as well as in batch normalization and layer normalization method. This was done in order to prevent the methods from overtly violating the Lipschitz condition. When we experimented *with* different multiplier parameter, we were in fact not able to achieve any improvement.

For optimization, we used the Adam optimizer Kingma & Ba (2015) in all of our experiments. We tested with 6 settings for (1) $n_{\mathrm{dis}}$, the number of updates of the discriminator per one update of the generator and (2) learning rate $\alpha$ and the first and second order momentum parameters $(\beta_1, \beta_2)$ of Adam. We list the details of these settings in Table 1 in the appendix section. Out of these 6 settings, A, B, and C are the settings used in previous representative works. The purpose of the settings D, E, and F is to the evaluate the performance of the algorithms implemented with more aggressive learning rates. For the details of the architectures of convolutional networks deployed in the generator and the discriminator, we refer the readers to Table 3 in the appendix section. The number of updates for GAN generator were 100K for all experiments, unless otherwise noted.

Firstly, we inspected the spectral norm of each layer during the training to make sure that our spectral normalization procedure is indeed serving its purpose. As we can see in the Figure 9 in the C.1,

Table 1: Hyper-parameter settings we tested in our experiments. †, ‡ and ⋆ are the hyperparameter settings following Gulrajani et al. (2017), Warde-Farley & Bengio (2017) and Radford et al. (2016), respectively.

| Setting | $\alpha$ | $\beta_1$ | $\beta_2$ | $n_{\mathrm{dis}}$ |
|---|---|---|---|---|
| A† | 0.0001 | 0.5 | 0.9 | 5 |
| B‡ | 0.0001 | 0.5 | 0.999 | 1 |
| C⋆ | 0.0002 | 0.5 | 0.999 | 1 |
| D | 0.001 | 0.5 | 0.9 | 5 |
| E | 0.001 | 0.5 | 0.999 | 5 |
| F | 0.001 | 0.9 | 0.999 | 5 |

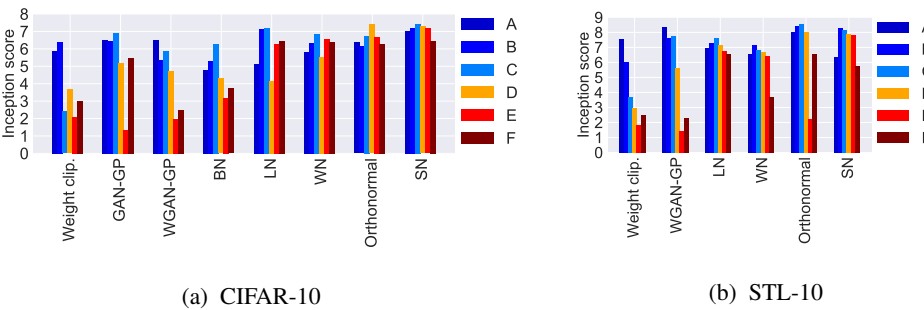

(a) CIFAR-10      (b) STL-10

Figure 1: Inception scores on CIFAR-10 and STL-10 with different methods and hyperparameters (higher is better).

the spectral norms of these layers floats around 1–1.05 region throughout the training. Please see Appendix C.1 for more details.

In Figures 1 and 2 we show the inception scores of each method with the settings A–F. We can see that spectral normalization is relatively robust with aggressive learning rates and momentum parameters. WGAN-GP fails to train good GANs at high learning rates and high momentum parameters on both CIFAR-10 and STL-10. Orthonormal regularization performed poorly for the setting E on the STL-10, but performed slightly better than our method with the optimal setting. These results suggests that our method is more robust than other methods with respect to the change in the setting of the training. Also, the optimal performance of weight normalization was inferior to both WGAN-GP and spectral normalization on STL-10, which consists of more diverse examples than CIFAR-10. Best scores of spectral normalization are better than almost all other methods on both CIFAR-10 and STL-10.

In Tables 2, we show the inception scores of the different methods with optimal settings on CIFAR-10 and STL-10 dataset. We see that SN-GANs performed better than almost all contemporaries on the optimal settings. SN-GANs performed even better with hinge loss (17).[4]. For the training with same number of iterations, SN-GANs fell behind orthonormal regularization for STL-10. For more detailed comparison between orthonormal regularization and spectral normalization, please see section 4.1.2.

In Figure 6 we show the images produced by the generators trained with WGAN-GP, weight normalization, and spectral normalization. SN-GANs were consistently better than GANs with weight normalization in terms of the quality of generated images. To be more precise, as we mentioned in Section 3, the set of images generated by spectral normalization was clearer and more diverse than the images produced by the weight normalization. We can also see that WGAN-GP failed to train good GANs with high learning rates and high momentums (D,E and F). The generated images

---

[4]As for STL-10, we also ran SN-GANs over twice time longer iterations because it did not seem to converge. Yet still, this elongated training sequence still completes before WGAN-GP with original iteration size because the optimal setting of SN-GANs (setting B, $n_{dis} = 1$) is computationally light.

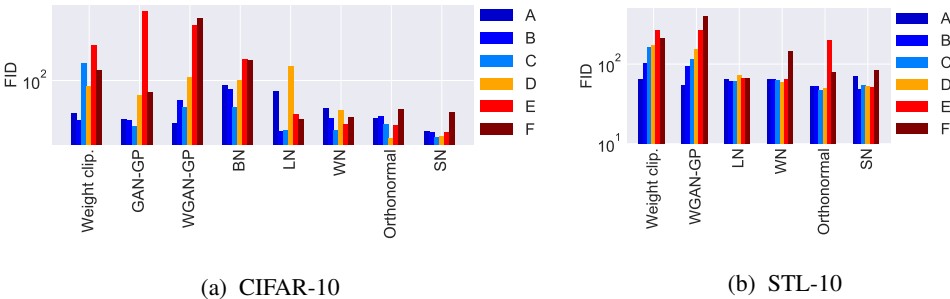

(a) CIFAR-10        (b) STL-10

Figure 2: FIDs on CIFAR-10 and STL-10 with different methods and hyperparameters (lower is better).

Table 2: Inception scores and FIDs with unsupervised image generation on CIFAR-10. † (Radford et al., 2016) (experimented by Yang et al. (2017)), ‡ (Yang et al., 2017), ∗ (Warde-Farley & Bengio, 2017), †† (Gulrajani et al., 2017)

| Method | Inception score | | FID | |
|---|---|---|---|---|
| | CIFAR-10 | STL-10 | CIFAR-10 | STL-10 |
| Real data | 11.24±.12 | 26.08±.26 | 7.8 | 7.9 |
| **-Standard CNN-** | | | | |
| Weight clipping | 6.41±.11 | 7.57±.10 | 42.6 | 64.2 |
| GAN-GP | 6.93±.08 | | 37.7 | |
| WGAN-GP | 6.68±.06 | 8.42±.13 | 40.2 | 55.1 |
| Batch Norm. | 6.27±.10 | | 56.3 | |
| Layer Norm. | 7.19±.12 | 7.61±.12 | 33.9 | 75.6 |
| Weight Norm. | 6.84±.07 | 7.16±.10 | 34.7 | 73.4 |
| Orthonormal | 7.40±.12 | 8.56±.07 | 29.0 | 46.7 |
| (ours) SN-GANs | 7.42±.08 | 8.28±.09 | 29.3 | 53.1 |
| Orthonormal (2x updates) | | 8.67±.08 | | 44.2 |
| (ours) SN-GANs (2x updates) | | 8.69±.09 | | 47.5 |
| (ours) SN-GANs, Eq.(17) | 7.58±.12 | | 25.5 | |
| (ours) SN-GANs, Eq.(17) (2x updates) | | 8.79±.14 | | 43.2 |
| **-ResNet-**[5] | | | | |
| Orthonormal, Eq.(17) | 7.92±.04 | 8.72±.06 | 23.8±.58 | 42.4±.99 |
| (ours) SN-GANs, Eq.(17) | **8.22**±.05 | **9.10**±.04 | **21.7**±.21 | **40.1**±.50 |
| DCGAN† | 6.64±.14 | 7.84±.07 | | |
| LR-GANs‡ | 7.17±.07 | | | |
| Warde-Farley et al.∗ | 7.72±.13 | 8.51±.13 | | |
| WGAN-GP (ResNet)†† | 7.86±.08 | | | |

with GAN-GP, batch normalization, and layer normalization is shown in Figure 12 in the appendix section.

We also compared our algorithm against multiple benchmark methods ans summarized the results on the bottom half of the Table 2. We also tested the performance of our method on ResNet based GANs used in Gulrajani et al. (2017). Please note that all methods listed thereof are all different in both optimization methods and the architecture of the model. Please see Table 4 and 5 in the appendix section for the detail network architectures. Our implementation of our algorithm was able to perform better than almost all the predecessors in the performance.

---

[5]For our ResNet experiments, we trained the same architecture with multiple random seeds for weight initialization and produced models with different parameters. We then generated 5000 images 10 times and computed the average inception score for each model. The values for ResNet on the table are the mean and standard deviation of the score computed over the set of models trained with different seeds.

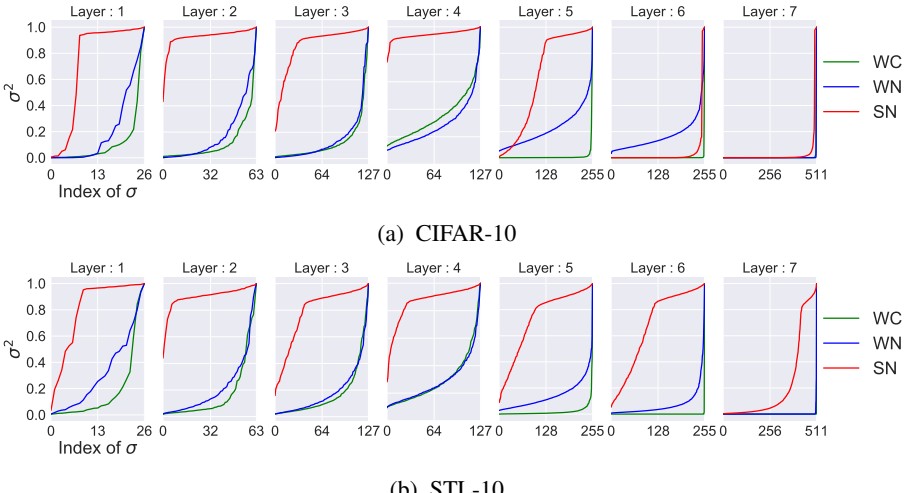

(a) CIFAR-10

(b) STL-10

Figure 3: Squared singular values of weight matrices trained with different methods: Weight clipping (WC), Weight Normalization (WN) and Spectral Normalization (SN). We scaled the singular values so that the largest singular values is equal to 1. For WN and SN, we calculated singular values of the normalized weight matrices.

### 4.1.1 ANALYSIS OF SN-GANS

**Singular values analysis on the weights of the discriminator** $D$   In Figure 3, we show the squared singular values of the weight matrices in the final discriminator $D$ produced by each method using the parameter that yielded the best inception score. As we predicted in Section 3, the singular values of the first to fifth layers trained with weight clipping and weight normalization concentrate on a few components. That is, the weight matrices of these layers tend to be rank deficit. On the other hand, the singular values of the weight matrices in those layers trained with spectral normalization is more broadly distributed. When the goal is to distinguish a pair of probability distributions on the low-dimensional nonlinear data manifold embedded in a high dimensional space, rank deficiencies in lower layers can be especially fatal. Outputs of lower layers have gone through only a few sets of rectified linear transformations, which means that they tend to lie on the space that is linear in most parts. Marginalizing out many features of the input distribution in such space can result in oversimplified discriminator. We can actually confirm the effect of this phenomenon on the generated images especially in Figure 6b. The images generated with spectral normalization is more diverse and complex than those generated with weight normalization.

**Training time**   On CIFAR-10, SN-GANs is slightly slower than weight normalization (about 110 $\sim$ 120% computational time), but significantly faster than WGAN-GP. As we mentioned in Section 3, WGAN-GP is slower than other methods because WGAN-GP needs to calculate the gradient of gradient norm $\|\nabla_x D\|_2$. For STL-10, the computational time of SN-GANs is almost the same as vanilla GANs, because the relative computational cost of the power iteration (18) is negligible when compared to the cost of forward and backward propagation on CIFAR-10 (images size of STL-10 is larger ($48 \times 48$)). Please see Figure 10 in the appendix section for the actual computational time.

### 4.1.2 COMPARISON BETWEEN SN-GANS AND ORTHONORMAL REGULARIZATION

In order to highlight the difference between our spectral normalization and orthonormal regularization, we conducted an additional set of experiments. As we explained in Section 3, orthonormal regularization is different from our method in that it destroys the spectral information and puts equal emphasis on all feature dimensions, including the ones that 'shall' be weeded out in the training process. To see the extent of its possibly detrimental effect, we experimented by increasing the di-

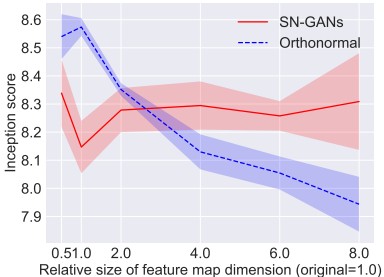

Figure 4: The effect on the performance on STL-10 induced by the change of the feature map dimension of the final layer. The width of the highlighted region represents standard deviation of the results over multiple seeds of weight initialization. The *orthonormal regularization* does not perform well with large feature map dimension, possibly because of its design that forces the discriminator to use all dimensions including the ones that are unnecessary. For the setting of the optimizers' hyper-parameters, We used the setting C, which was optimal for "*orthonormal regularization*"

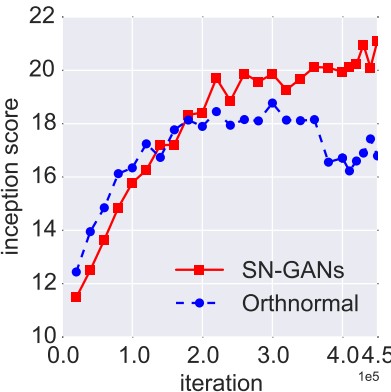

Figure 5: Learning curves for conditional image generation in terms of Inception score for SN-GANs and GANs with *orthonormal regularization* on ImageNet.

mension of the feature space [6], especially at the final layer (7th conv) for which the training with our spectral normalization prefers relatively small feature space (dimension $< 100$; see Figure 3b). As for the setting of the training, we selected the parameters for which the orthonormal regularization performed optimally. The figure 4 shows the result of our experiments. As we predicted, the performance of the orthonormal regularization deteriorates as we increase the dimension of the feature maps at the final layer. Our SN-GANs, on the other hand, does not falter with this modification of the architecture. Thus, at least in this perspective, we may such that our method is more robust with respect to the change of the network architecture.

## 4.2 IMAGE GENERATION ON IMAGENET

To show that our method remains effective on a large high dimensional dataset, we also applied our method to the training of conditional GANs on ILRSVRC2012 dataset with 1000 classes, each consisting of approximately 1300 images, which we compressed to $128 \times 128$ pixels. Regarding the adversarial loss for conditional GANs, we used practically the same formulation used in Mirza & Osindero (2014), except that we replaced the standard GANs loss with hinge loss (17). Please see Appendix B.3 for the details of experimental settings.

---

[6]More precisely, we simply increased the input dimension and the output dimension by the same factor. In Figure 4, 'relative size' = 1.0 implies that the layer structure is the same as the original.

GANs without normalization and GANs with layer normalization collapsed in the beginning of training and failed to produce any meaningful images. GANs with orthonormal normalization Brock et al. (2016) and our spectral normalization, on the other hand, was able to produce images. The inception score of the orthonormal normalization however plateaued around 20Kth iterations, while SN kept improving even afterward (Figure 5.) To our knowledge, our research is the first of its kind in succeeding to produce decent images from ImageNet dataset with a *single* pair of a discriminator and a generator (Figure 7). To measure the degree of mode-collapse, we followed the footstep of Odena et al. (2017) and computed the intra MS-SSIM Odena et al. (2017) for pairs of independently generated GANs images of each class. We see that our SN-GANs ((intra MS-SSIM)=0.101) is suffering less from the mode-collapse than AC-GANs ((intra MS-SSIM)∼0.25).

To ensure that the superiority of our method is not limited within our specific setting, we also compared the performance of SN-GANs against *orthonormal regularization* on conditional GANs with *projection discriminator* (Miyato & Koyama, 2018) as well as the standard (unconditional) GANs. In our experiments, SN-GANs achieved better performance than *orthonormal regularization* for the both settings (See Figure 13 in the appendix section).

## 5    CONCLUSION

This paper proposes spectral normalization as a stabilizer of training of GANs. When we apply spectral normalization to the GANs on image generation tasks, the generated examples are more diverse than the conventional weight normalization and achieve better or comparative inception scores relative to previous studies. The method imposes global regularization on the discriminator as opposed to local regularization introduced by WGAN-GP, and can possibly used in combinations. In the future work, we would like to further investigate where our methods stand amongst other methods on more theoretical basis, and experiment our algorithm on larger and more complex datasets.

### ACKNOWLEDGMENTS

We would like to thank the members of Preferred Networks, Inc., particularly Shin-ichi Maeda, Eiichi Matsumoto, Masaki Watanabe and Keisuke Yahata for insightful comments and discussions. We also would like to thank anonymous reviewers and commenters on the OpenReview forum for insightful discussions.

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

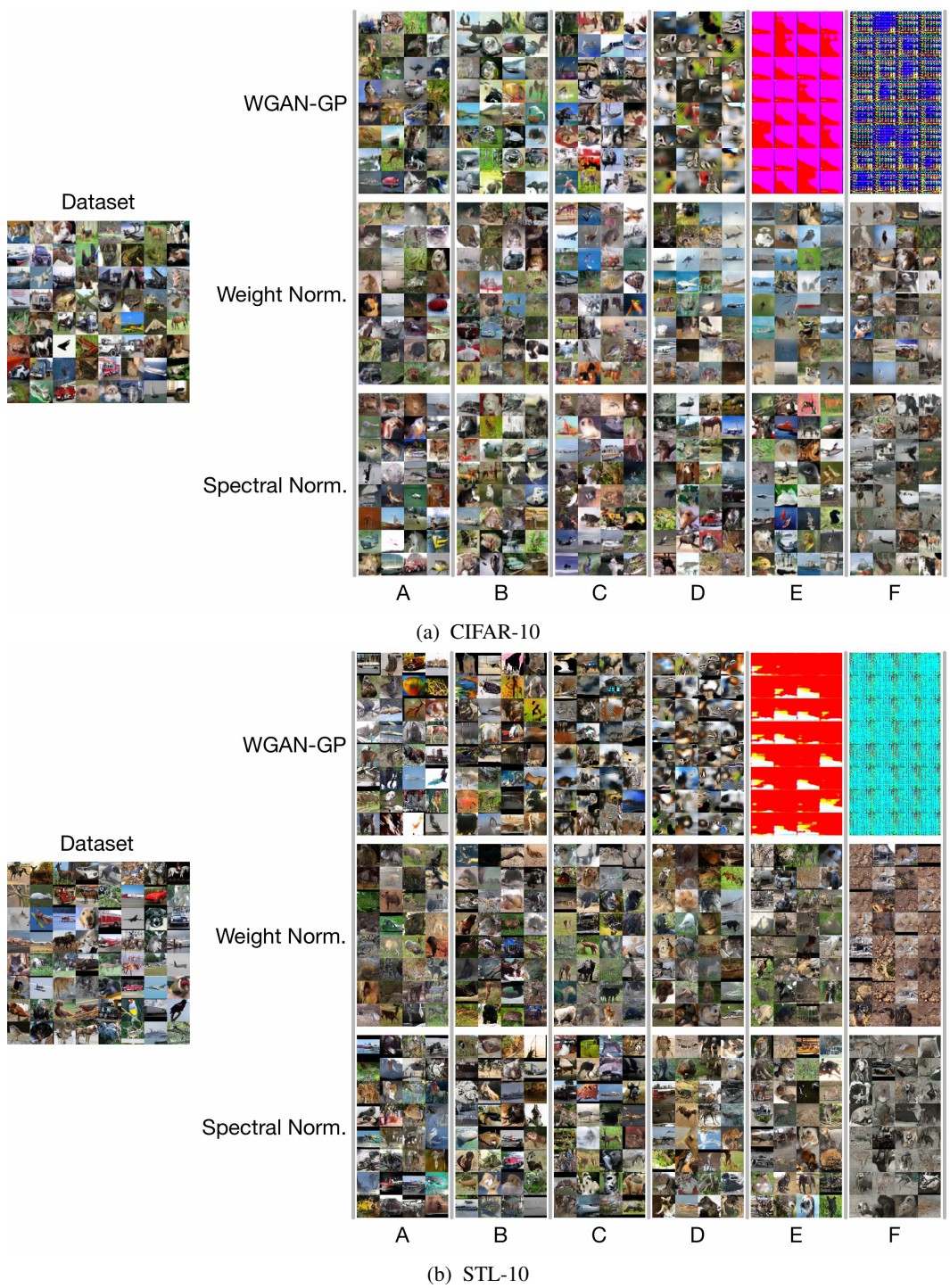

(a) CIFAR-10

(b) STL-10

Figure 6: Generated images on different methods: WGAN-GP, weight normalization, and spectral normalization on CIFAR-10 and STL-10.

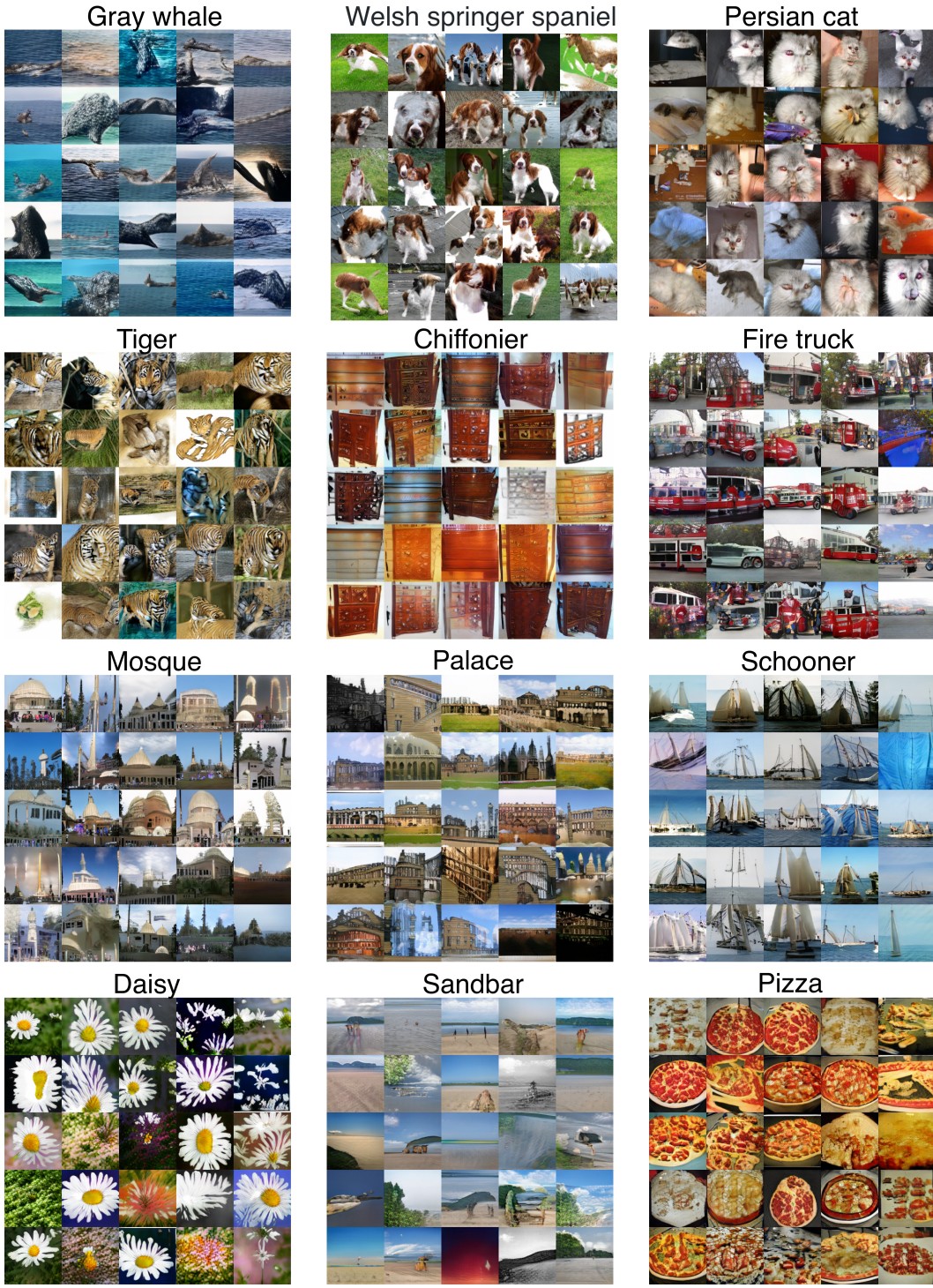

Figure 7: 128x128 pixel images generated by SN-GANs trained on ILSVRC2012 dataset. The inception score is 21.1±.35.

# A   THE ALGORITHM OF SPECTRAL NORMALIZATION

Let us describe the shortcut in Section 2.1 in more detail. We begin with vectors $\tilde{u}$ that is randomly initialized for each weight. If there is no multiplicity in the dominant singular values and if $\tilde{u}$ is not orthogonal to the first left singular vectors[7], we can appeal to the principle of the power method and produce the first left and right singular vectors through the following update rule:

$$\tilde{v} \leftarrow W^{\mathrm{T}}\tilde{u}/\|W^{\mathrm{T}}\tilde{u}\|_2, \; \tilde{u} \leftarrow W\tilde{v}/\|W\tilde{v}\|_2. \tag{18}$$

We can then approximate the spectral norm of $W$ with the pair of so-approximated singular vectors:

$$\sigma(W) \approx \tilde{u}^{\mathrm{T}}W\tilde{v}. \tag{19}$$

If we use SGD for updating $W$, the change in $W$ at each update would be small, and hence the change in its largest singular value. In our implementation, we took advantage of this fact and reused the $\tilde{u}$ computed at each step of the algorithm as the initial vector in the subsequent step. In fact, with this 'recycle' procedure, one round of power iteration was sufficient in the actual experiment to achieve satisfactory performance. Algorithm 1 in the appendix summarizes the computation of the spectrally normalized weight matrix $\bar{W}$ with this approximation. Note that this procedure is very computationally cheap even in comparison to the calculation of the forward and backward propagations on neural networks. Please see Figure 10 for actual computational time with and without spectral normalization.

---

**Algorithm 1** SGD with spectral normalization

---

- Initialize $\tilde{u}_l \in \mathcal{R}^{d_l}$ for $l = 1, \ldots, L$ with a random vector (sampled from isotropic distribution).
- For each update and each layer $l$:
    1. Apply power iteration method to a unnormalized weight $W^l$:

    $$\tilde{v}_l \leftarrow (W^l)^{\mathrm{T}}\tilde{u}_l/\|(W^l)^{\mathrm{T}}\tilde{u}_l\|_2 \tag{20}$$

    $$\tilde{u}_l \leftarrow W^l\tilde{v}_l/\|W^l\tilde{v}_l\|_2 \tag{21}$$

    2. Calculate $\bar{W}_{\mathrm{SN}}$ with the spectral norm:

    $$\bar{W}^l_{\mathrm{SN}}(W^l) = W^l/\sigma(W^l), \; \text{where } \sigma(W^l) = \tilde{u}_l^{\mathrm{T}}W^l\tilde{v}_l \tag{22}$$

    3. Update $W^l$ with SGD on mini-batch dataset $\mathcal{D}_M$ with a learning rate $\alpha$:

    $$W^l \leftarrow W^l - \alpha\nabla_{W^l}\ell(\bar{W}^l_{\mathrm{SN}}(W^l), \mathcal{D}_M) \tag{23}$$

---

# B   EXPERIMENTAL SETTINGS

## B.1   PERFORMANCE MEASURES

Inception score is introduced originally by Salimans et al. (2016):  $I(\{x_n\}_{n=1}^N) := \exp(\mathrm{E}[D_{\mathrm{KL}}[p(y|\boldsymbol{x})\|p(y)]])$, where $p(y)$ is approximated by $\frac{1}{N}\sum_{n=1}^N p(y|\boldsymbol{x}_n)$ and $p(y|x)$ is the trained Inception convolutional neural network (Szegedy et al., 2015), which we would refer to as Inception model for short. In their work, Salimans et al. (2016) reported that this score is strongly correlated with subjective human judgment of image quality. Following the procedure in Salimans et al. (2016); Warde-Farley & Bengio (2017), we calculated the score for randomly generated 5000 examples from each trained generator to evaluate its ability to generate natural images. We repeated each experiment 10 times and reported the average and the standard deviation of the inception scores.

Fréchet inception distance (Heusel et al., 2017) is another measure for the quality of the generated examples that uses 2nd order information of the final layer of the inception model applied to the

---

[7]In practice, we are safe to assume that $\tilde{u}$ generated from uniform distribution on the sphere is not orthogonal to the first singular vectors, because this can happen with probability 0.

examples. On its own, the *Frećhet distance* Dowson & Landau (1982) is 2-Wasserstein distance between two distribution $p_1$ and $p_2$ assuming they are both multivariate Gaussian distributions:

$$F(p_1, p_2) = \|\boldsymbol{\mu}_{p_1} - \boldsymbol{\mu}_{p_2}\|_2^2 + \text{trace}\left(C_{p_1} + C_{p_2} - 2(C_{p_1} C_{p_2})^{1/2}\right), \tag{24}$$

where $\{\boldsymbol{\mu}_{p_1}, C_{p_1}\}$, $\{\boldsymbol{\mu}_{p_2}, C_{p_2}\}$ are the mean and covariance of samples from $q$ and $p$, respectively. If $f_{\ominus}$ is the output of the final layer of the inception model before the softmax, the Fréchet inception distance (FID) between two distributions $p_1$ and $p_2$ on the images is the distance between $f_{\ominus} \circ p_1$ and $f_{\ominus} \circ p_2$. We computed the Fréchet inception distance between the true distribution and the generated distribution empirically over 10000 and 5000 samples. Multiple repetition of the experiments did not exhibit any notable variations on this score.

## B.2 IMAGE GENERATION ON CIFAR-10 AND STL-10

For the comparative study, we experimented with the recent ResNet architecture of Gulrajani et al. (2017) as well as the standard CNN. For this additional set of experiments, we used Adam again for the optimization and used the very hyper parameter used in Gulrajani et al. (2017) ($\alpha = 0.0002, \beta_1 = 0, \beta_2 = 0.9, n_{dis} = 5$). For our SN-GANs, we doubled the feature map in the generator from the original, because this modification achieved better results. Note that when we doubled the dimension of the feature map for the WGAN-GP experiment, however, the performance deteriorated.

## B.3 IMAGE GENERATION ON IMAGENET

The images used in this set of experiments were resized to $128 \times 128$ pixels. The details of the architecture are given in Table 6. For the generator network of conditional GANs, we used conditional batch normalization (CBN) (Dumoulin et al., 2017; de Vries et al., 2017). Namely we replaced the standard batch normalization layer with the CBN conditional to the label information $y \in \{1, \ldots, 1000\}$. For the optimization, we used Adam with the same hyperparameters we used for ResNet on CIFAR-10 and STL-10 dataset. We trained the networks with 450K generator updates, and applied linear decay for the learning rate after 400K iterations so that the rate would be 0 at the end.

## B.4 NETWORK ARCHITECTURES

Table 3: Standard CNN models for CIFAR-10 and STL-10 used in our experiments on image Generation. The slopes of all lReLU functions in the networks are set to 0.1.

| $z \in \mathbb{R}^{128} \sim \mathcal{N}(0, I)$ |
| --- |
| dense $\to M_g \times M_g \times 512$ |
| $4 \times 4$, stride=2 deconv. BN 256 ReLU |
| $4 \times 4$, stride=2 deconv. BN 128 ReLU |
| $4 \times 4$, stride=2 deconv. BN 64 ReLU |
| $3 \times 3$, stride=1 conv. 3 Tanh |

(a) Generator, $M_g = 4$ for SVHN and CIFAR10, and $M_g = 6$ for STL-10

| RGB image $x \in \mathbb{R}^{M \times M \times 3}$ |
| --- |
| $3 \times 3$, stride=1 conv 64 lReLU |
| $4 \times 4$, stride=2 conv 64 lReLU |
| $3 \times 3$, stride=1 conv 128 lReLU |
| $4 \times 4$, stride=2 conv 128 lReLU |
| $3 \times 3$, stride=1 conv 256 lReLU |
| $4 \times 4$, stride=2 conv 256 lReLU |
| $3 \times 3$, stride=1 conv. 512 lReLU |
| dense $\to 1$ |

(b) Discriminator, $M = 32$ for SVHN and CIFAR10, and $M = 48$ for STL-10

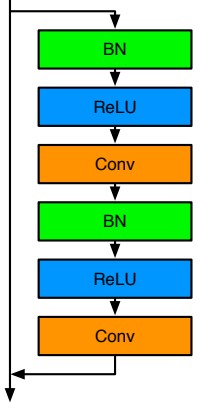

Table 4: ResNet architectures for CIFAR10 dataset. We use similar architectures to the ones used in Gulrajani et al. (2017).

Figure 8: ResBlock architecture. For the discriminator we removed BN layers in ResBlock.

| $z \in \mathbb{R}^{128} \sim \mathcal{N}(0, I)$ |
| --- |
| dense, $4 \times 4 \times 256$ |
| ResBlock up 256 |
| ResBlock up 256 |
| ResBlock up 256 |
| BN, ReLU, 3×3 conv, 3 Tanh |
| (a) Generator |

| RGB image $x \in \mathbb{R}^{32 \times 32 \times 3}$ |
| --- |
| ResBlock down 128 |
| ResBlock down 128 |
| ResBlock 128 |
| ResBlock 128 |
| ReLU |
| Global sum pooling |
| dense $\to 1$ |
| (b) Discriminator |

Table 5: ResNet architectures for STL-10 dataset.

| $z \in \mathbb{R}^{128} \sim \mathcal{N}(0, I)$ |
| --- |
| dense, $6 \times 6 \times 512$ |
| ResBlock up 256 |
| ResBlock up 128 |
| ResBlock up 64 |
| BN, ReLU, 3×3 conv, 3 Tanh |
| (a) Generator |

| RGB image $x \in \mathbb{R}^{48 \times 48 \times 3}$ |
| --- |
| ResBlock down 64 |
| ResBlock down 128 |
| ResBlock down 256 |
| ResBlock down 512 |
| ResBlock 1024 |
| ReLU |
| Global sum pooling |
| dense $\to 1$ |
| (b) Discriminator |

Table 6: ResNet architectures for image generation on ImageNet dataset. For the generator of conditional GANs, we replaced the usual batch normalization layer in the ResBlock with the conditional batch normalization layer. As for the model of the *projection discriminator*, we used the same architecture used in Miyato & Koyama (2018). Please see the paper for the details.

| $z \in \mathbb{R}^{128} \sim \mathcal{N}(0, I)$ |
| --- |
| dense, $4 \times 4 \times 1024$ |
| ResBlock up 1024 |
| ResBlock up 512 |
| ResBlock up 256 |
| ResBlock up 128 |
| ResBlock up 64 |
| BN, ReLU, $3\times3$ conv 3 |
| Tanh |

(a) Generator

| RGB image $x \in \mathbb{R}^{128\times128\times3}$ |
| --- |
| ResBlock down 64 |
| ResBlock down 128 |
| ResBlock down 256 |
| ResBlock down 512 |
| ResBlock down 1024 |
| ResBlock 1024 |
| ReLU |
| Global sum pooling |
| dense $\rightarrow$ 1 |

(b) Discriminator for unconditional GANs.

| RGB image $x \in \mathbb{R}^{128\times128\times3}$ |
| --- |
| ResBlock down 64 |
| ResBlock down 128 |
| ResBlock down 256 |
| Concat(Embed($y$), $\boldsymbol{h}$) |
| ResBlock down 512 |
| ResBlock down 1024 |
| ResBlock 1024 |
| ReLU |
| Global sum pooling |
| dense $\rightarrow$ 1 |

(c) Discriminator for conditional GANs. For computational ease, we embedded the integer label $y \in \{0, \ldots, 1000\}$ into 128 dimension before concatenating the vector to the output of the intermediate layer.

## C  APPENDIX RESULTS

### C.1  ACCURACY OF SPECTRAL NORMALIZATION

Figure 9 shows the spectral norm of each layer in the discriminator over the course of the training. The setting of the optimizer is C in Table 1 throughout the training. In fact, they do not deviate by more than 0.05 for the most part. As an exception, 6 and 7-th convolutional layers with largest rank deviate by more than 0.1 in the beginning of the training, but the norm of this layer too stabilizes around 1 after some iterations.

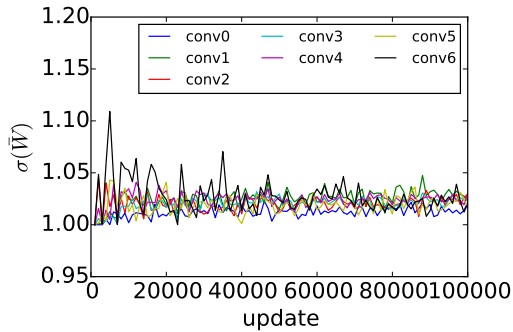

Figure 9:  Spectral norms of all seven convolutional layers in the standard CNN during course of the training on CIFAR 10.

### C.2  TRAINING TIME

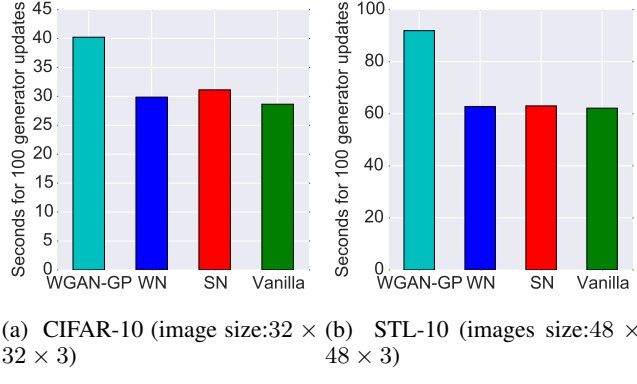

(a)  CIFAR-10 (image size:$32 \times$ (b)  STL-10 (images size:$48 \times$
$32 \times 3$)                          $48 \times 3$)

Figure 10: Computational time for 100 updates. We set $n_{\mathrm{dis}} = 5$

### C.3  THE EFFECT OF $n_{dis}$ ON SPECTRAL NORMALIZATION AND WEIGHT NORMALIZATION

Figure 11 shows the effect of $n_{dis}$ on the performance of weight normalization and spectral normalization. All results shown in Figure 11 follows setting D, except for the value of $n_{dis}$. For WN, the performance deteriorates with larger $n_{dis}$, which amounts to computing minimax with better accuracy. Our SN does not suffer from this unintended effect.

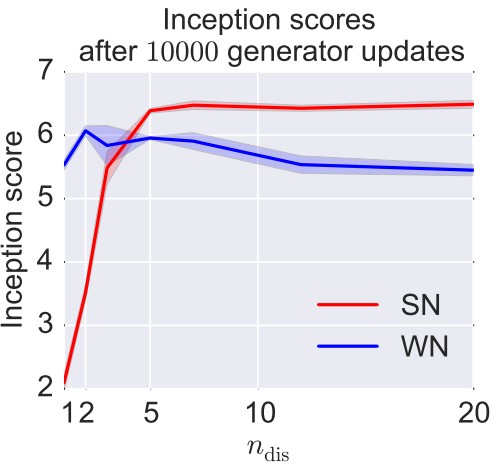

Figure 11: The effect of $n_{dis}$ on spectral normalization and weight normalization. The shaded region represents the variance of the result over different seeds.

### C.4 GENERATED IMAGES ON CIFAR10 WITH GAN-GP, LAYER NORMALIZATION AND BATCH NORMALIZATION

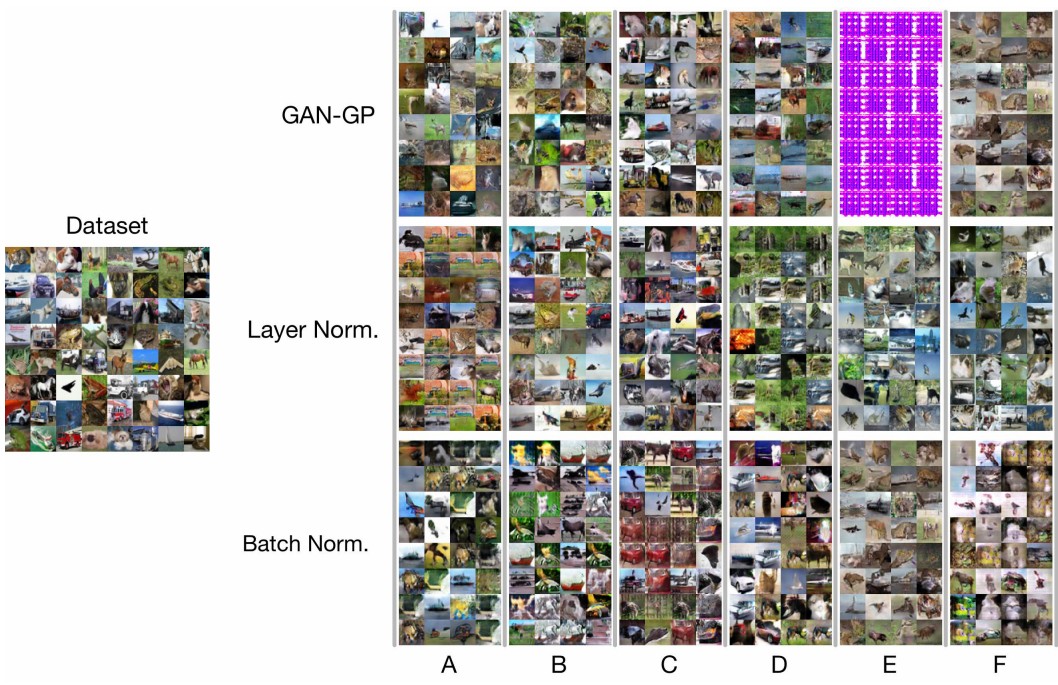

Figure 12: Generated images with GAN-GP, Layer Norm and Batch Norm on CIFAR-10

## C.5 IMAGE GENERATION ON IMAGENET

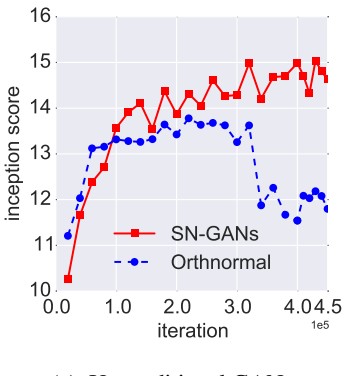
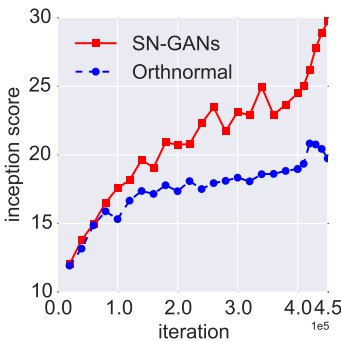

(a) Unconditional GANs

(b) Conditional GANs with *projection discriminator*

Figure 13: Learning curves in terms of Inception score for SN-GANs and GANs with *orthonormal regularization* on ImageNet. The figure (a) shows the results for the standard (unconditional) GANs, and the figure (b) shows the results for the conditional GANs trained with *projection discriminator* (Miyato & Koyama, 2018)

## D SPECTRAL NORMALIZATION VS OTHER REGULARIZATION TECHNIQUES

This section is dedicated to the comparative study of spectral normalization and other regularization methods for discriminators. In particular, we will show that contemporary regularizations including weight normalization and weight clipping implicitly impose constraints on weight matrices that places unnecessary restriction on the search space of the discriminator. More specifically, we will show that weight normalization and weight clipping *unwittingly* favor low-rank weight matrices. This can force the trained discriminator to be largely dependent on select few features, rendering the algorithm to be able to match the model distribution with the target distribution only on very low dimensional feature space.

### D.1 WEIGHT NORMALIZATION AND FROBENIUS NORMALIZATION

The weight normalization introduced by Salimans & Kingma (2016) is a method that normalizes the $\ell_2$ norm of each row vector in the weight matrix[8]:

$$\bar{W}_{\mathrm{WN}} := \left[\bar{\boldsymbol{w}}_1^{\mathrm{T}}, \bar{\boldsymbol{w}}_2^{\mathrm{T}}, ..., \bar{\boldsymbol{w}}_{d_o}^{\mathrm{T}}\right]^{\mathrm{T}}, \text{ where } \bar{\boldsymbol{w}}_i(\boldsymbol{w}_i) := \boldsymbol{w}_i/\|\boldsymbol{w}_i\|_2, \tag{25}$$

where $\bar{\boldsymbol{w}}_i$ and $\boldsymbol{w}_i$ are the $i$th row vector of $\bar{W}_{\mathrm{WN}}$ and $W$, respectively.

Still another technique to regularize the weight matrix is to use the Frobenius norm:

$$\bar{W}_{\mathrm{FN}} := W/\|W\|_F, \tag{26}$$

where $\|W\|_F := \sqrt{\mathrm{tr}(W^{\mathrm{T}}W)} = \sqrt{\sum_{i,j} w_{ij}^2}$.

Originally, these regularization techniques were invented with the goal of improving the generalization performance of supervised training (Salimans & Kingma, 2016; Arpit et al., 2016). However, recent works in the field of GANs (Salimans et al., 2016; Xiang & Li, 2017) found their another raison d'etat as a regularizer of discriminators, and succeeded in improving the performance of the original.

---

[8]In the original literature, the weight normalization was introduced as a method for reparametrization of the form $\bar{W}_{\mathrm{WN}} := \left[\gamma_1 \bar{\boldsymbol{w}}_1^{\mathrm{T}}, \gamma_2 \bar{\boldsymbol{w}}_2^{\mathrm{T}}, ..., \gamma_{d_o} \bar{\boldsymbol{w}}_{d_o}^{\mathrm{T}}\right]^{\mathrm{T}}$ where $\gamma_i \in \mathbb{R}$ is to be learned in the course of the training. In this work, we deal with the case $\gamma_i = 1$ so that we can assess the methods under the Lipschitz constraint.

These methods in fact can render the trained discriminator $D$ to be $K$-Lipschitz for a some prescribed $K$ and achieve the desired effect to a certain extent. However, weight normalization (25) imposes the following implicit restriction on the choice of $\bar{W}_{\mathrm{WN}}$:

$$\sigma_1(\bar{W}_{\mathrm{WN}})^2 + \sigma_2(\bar{W}_{\mathrm{WN}})^2 + \cdots + \sigma_T(\bar{W}_{\mathrm{WN}})^2 = d_o, \text{ where } T = \min(d_i, d_o), \qquad (27)$$

where $\sigma_t(A)$ is a $t$-th singular value of matrix $A$. The above equation holds because $\sum_{t=1}^{\min(d_i,d_o)} \sigma_t(\bar{W}_{\mathrm{WN}})^2 = \mathrm{tr}(\bar{W}_{\mathrm{WN}}\bar{W}_{\mathrm{WN}}^{\mathrm{T}}) = \sum_{i=1}^{d_o} \frac{\boldsymbol{w}_i}{\|\boldsymbol{w}_i\|_2} \frac{\boldsymbol{w}_i^{\mathrm{T}}}{\|\boldsymbol{w}_i\|_2} = d_o$. Under this restriction, the norm $\|\bar{W}_{\mathrm{WN}}\boldsymbol{h}\|_2$ for a fixed unit vector $\boldsymbol{h}$ is maximized at $\|\bar{W}_{\mathrm{WN}}\boldsymbol{h}\|_2 = \sqrt{d_o}$ when $\sigma_1(\bar{W}_{\mathrm{WN}}) = \sqrt{d_o}$ and $\sigma_t(\bar{W}_{\mathrm{WN}}) = 0$ for $t = 2, \ldots, T$, which means that $\bar{W}_{\mathrm{WN}}$ is of rank one. Using such $W$ corresponds to using only one feature to discriminate the model probability distribution from the target. Similarly, Frobenius normalization requires $\sigma_1(\bar{W}_{\mathrm{FN}})^2 + \sigma_2(\bar{W}_{\mathrm{FN}})^2 + \cdots + \sigma_T(\bar{W}_{\mathrm{FN}})^2 = 1$, and the same argument as above follows.

Here, we see a critical problem in these two regularization methods. In order to retain as much norm of the input as possible and hence to make the discriminator more sensitive, one would hope to make the norm of $\bar{W}_{\mathrm{WN}}\boldsymbol{h}$ large. For weight normalization, however, this comes at the cost of reducing the rank and hence the number of features to be used for the discriminator. Thus, there is a conflict of interests between weight normalization and our desire to use as many features as possible to distinguish the generator distribution from the target distribution. The former interest often reigns over the other in many cases, inadvertently diminishing the number of features to be used by the discriminators. Consequently, the algorithm would produce a rather arbitrary model distribution that matches the target distribution only at select few features.

Our spectral normalization, on the other hand, do not suffer from such a conflict in interest. Note that the Lipschitz constant of a linear operator is determined only by the maximum singular value. In other words, the spectral norm is independent of rank. Thus, unlike the weight normalization, our spectral normalization allows the parameter matrix to use as many features as possible while satisfying local 1-Lipschitz constraint. Our spectral normalization leaves more freedom in choosing the number of singular components (features) to feed to the next layer of the discriminator.

To see this more visually, we refer the reader to Figure (14). Note that spectral normalization allows for a wider range of choices than weight normalization.

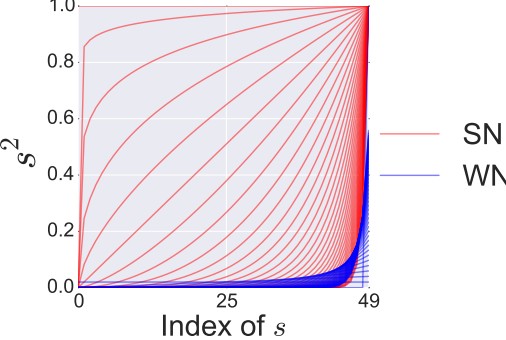

Figure 14: **Visualization of the difference between spectral normalization (Red) and weight normalization (Blue) on possible sets of singular values.** The possible sets of singular values plotted in increasing order for weight normalization (Blue) and for spectral normalization (Red). For the set of singular values permitted under the spectral normalization condition, we scaled $\bar{W}_{\mathrm{WN}}$ by $1/\sqrt{d_o}$ so that its spectral norm is exactly 1. By the definition of the weight normalization, *the area under the blue curves are all bound to be* 1. Note that the range of choice for the weight normalization is small.

In summary, weight normalization and Frobenius normalization favor skewed distributions of singular values, making the column spaces of the weight matrices lie in (approximately) low dimensional vector spaces. On the other hand, our spectral normalization does not compromise the number of feature dimensions used by the discriminator. In fact, we will experimentally show that GANs

trained with our spectral normalization can generate a synthetic dataset with wider variety and higher inception score than the GANs trained with other two regularization methods.

## D.2 WEIGHT CLIPPING

Still another regularization technique is *weight clipping* introduced by Arjovsky et al. (2017) in their training of Wasserstein GANs. Weight clipping simply truncates each element of weight matrices so that its absolute value is bounded above by a prescribed constant $c \in \mathbb{R}_+$. Unfortunately, weight clipping suffers from the same problem as weight normalization and Frobenius normalization. With weight clipping with the truncation value $c$, the value $\|Wx\|_2$ for a fixed unit vector $x$ is maximized when the rank of $W$ is again one, and the training will again favor the discriminators that use only select few features. Gulrajani et al. (2017) refers to this problem as capacity underuse problem. They also reported that the training of WGAN with weight clipping is slower than that of the original DCGAN (Radford et al., 2016).

## D.3 SINGULAR VALUE CLIPPING AND SINGULAR VALUE CONSTRAINT

One direct and straightforward way of controlling the spectral norm is to clip the singular values (Saito et al., 2017), (Jia et al., 2017). This approach, however, is computationally heavy because one needs to implement singular value decomposition in order to compute all the singular values.

A similar but less obvious approach is to parametrize $W \in \mathbb{R}^{d_o \times d_i}$ as follows from the get-go and train the discriminators with this constrained parametrization:

$$W := USV^{\mathrm{T}}, \ \ \text{subject to } U^{\mathrm{T}}U = I, V^{\mathrm{T}}V = I, \ \max_i S_{ii} = K, \tag{28}$$

where $U \in \mathbb{R}^{d_o \times P}$, $V \in \mathbb{R}^{d_i \times P}$, and $S \in \mathbb{R}^{P \times P}$ is a diagonal matrix. However, it is not a simple task to train this model while remaining absolutely faithful to this parametrization constraint. Our spectral normalization, on the other hand, can carry out the updates with relatively low computational cost without compromising the normalization constraint.

## D.4 WGAN WITH GRADIENT PENALTY (WGAN-GP)

Recently, Gulrajani et al. (2017) introduced a technique to enhance the stability of the training of Wasserstein GANs (Arjovsky et al., 2017). In their work, they endeavored to place $K$-Lipschitz constraint (5) on the discriminator by augmenting the adversarial loss function with the following regularizer function:

$$\lambda \mathop{\mathrm{E}}_{\hat{x} \sim p_{\hat{x}}} [(\|\nabla_{\hat{x}} D(\hat{x})\|_2 - 1)^2], \tag{29}$$

where $\lambda > 0$ is a balancing coefficient and $\hat{x}$ is:

$$\hat{x} := \epsilon x + (1 - \epsilon)\tilde{x} \tag{30}$$
$$\text{where } \epsilon \sim U[0, 1], \ x \sim p_{\mathrm{data}}, \ \tilde{x} = G(z), \ z \sim p_z. \tag{31}$$

Using this augmented objective function, Gulrajani et al. (2017) succeeded in training a GAN based on ResNet (He et al., 2016) with an impressive performance. The advantage of their method in comparison to spectral normalization is that they can impose local 1-Lipschitz constraint directly on the discriminator function without a rather round-about layer-wise normalization. This suggest that their method is less likely to underuse the capacity of the network structure.

At the same time, this type of method that penalizes the gradients at sample points $\hat{x}$ suffers from the obvious problem of not being able to regularize the function at the points outside of the support of the current generative distribution. In fact, the generative distribution and its support gradually changes in the course of the training, and this can destabilize the effect of the regularization itself.

On the contrary, our spectral normalization regularizes the function itself, and the effect of the regularization is more stable with respect to the choice of the batch. In fact, we observed in the experiment that a high learning rate can destabilize the performance of WGAN-GP. Training with our spectral normalization does not falter with aggressive learning rate.

Moreover, WGAN-GP requires more computational cost than our spectral normalization with single-step power iteration, because the computation of $\|\nabla_x D\|_2$ requires one whole round of forward and backward propagation. In Figure 10, we compare the computational cost of the two methods for the same number of updates.

Having said that, one shall not rule out the possibility that the gradient penalty can compliment spectral normalization and vice versa. Because these two methods regularizes discriminators by completely different means, and in the experiment section, we actually confirmed that combination of WGAN-GP and reparametrization with spectral normalization improves the quality of the generated examples over the baseline (WGAN-GP only).

## E    REPARAMETRIZATION MOTIVATED BY THE SPECTRAL NORMALIZATION

We can take advantage of the regularization effect of the spectral normalization we saw above to develop another algorithm. Let us consider another parametrization of the weight matrix of the discriminator given by:

$$\tilde{W} := \gamma \bar{W}_{\text{SN}} \tag{32}$$

where $\gamma$ is a scalar variable to be learned. This parametrization compromises the 1-Lipschitz constraint at the layer of interest, but gives more freedom to the model while keeping the model from becoming degenerate. For this reparametrization, we need to control the Lipschitz condition by other means, such as the gradient penalty (Gulrajani et al., 2017). Indeed, we can think of analogous versions of reparametrization by replacing $\bar{W}_{\text{SN}}$ in (32) with $W$ normalized by other criterions. The extension of this form is not new. In Salimans & Kingma (2016), they originally introduced weight normalization in order to derive the reparametrization of the form (32) with $\bar{W}_{\text{SN}}$ replaced (32) by $W_{\text{WN}}$ and vectorized $\gamma$.

### E.1    EXPERIMENTS: COMPARISON OF REPARAMETRIZATION WITH DIFFERENT NORMALIZATION METHODS

In this part of the addendum, we experimentally compare the reparametrizations derived from two different normalization methods (weight normalization and spectral normalization). We tested the reprametrization methods for the training of the discriminator of WGAN-GP. For the architecture of the network in WGAN-GP, we used the same CNN we used in the previous section. For the ResNet-based CNN, we used the same architecture provided by (Gulrajani et al., 2017) [9].

Tables 7, 8 summarize the result. We see that our method significantly improves the inception score from the baseline on the regular CNN, and slightly improves the score on the ResNet based CNN.

Figure 15 shows the learning curves of (a) critic losses, on train and validation sets and (b) the inception scores with different reparametrization methods. We can see the beneficial effect of spectral normalization in the learning curve of the discriminator as well. We can verify in the figure 15a that the discriminator with spectral normalization overfits less to the training dataset than the discriminator without reparametrization and with weight normalization, The effect of overfitting can be observed on inception score as well, and the final score with spectral normalization is better than the others. As for the best inception score achieved in the course of the training, spectral normalization achieved 7.28, whereas the spectral normalization and vanilla normalization achieved 7.04 and 6.69, respectively.

## F    THE GRADIENT OF GENERAL NORMALIZATION METHOD

Let us denote $\bar{W} := W/N(W)$ to be the normalized weight where $N(W)$ to be a scalar normalized coefficient (e.g. Spectral norm or Frobenius norm). In general, we can write the derivative of loss

---

[9]We implement our method based on the open-sourced code provided by the author (Gulrajani et al., 2017) https://github.com/igul222/improved_wgan_training/blob/master/gan_cifar_resnet.py

| Method | Inception score | FID |
|---|---|---|
| WGAN-GP (Standard CNN, Baseline) | 6.68±.06 | 40.1 |
| w/ Frobenius Norm. | N/A* | N/A* |
| w/ Weight Norm. | 6.36±.04 | 42.4 |
| w/ Spectral Norm. | **7.20±.08** | **32.0** |
| (WGAN-GP, ResNet, Gulrajani et al. (2017)) | 7.86±.08 | |
| WGAN-GP (ResNet, Baseline) | 7.80±.11 | 24.5 |
| w/ Spectral norm. | 7.85±.06 | 23.6 |
| w/ Spectral norm. (1.5x feature maps in $D$) | **7.96±.06** | **22.5** |

Table 7: Inception scores with different reparametrization mehtods on CIFAR10 without label supervisions. (*)We reported N/A for the inception score and FID of Frobenius normalization because the training collapsed at the early stage.

| Method (ResNet) | Inception score | FID |
|---|---|---|
| (AC-WGAN-GP, Gulrajani et al. (2017)) | 8.42±.10 | |
| AC-WGAN-GP (Baseline) | 8.29±.12 | 19.5 |
| w/ Spectral norm. | 8.59±.12 | 18.6 |
| w/ Spectral norm. (1.5x feature maps in $D$) | **8.60±.08** | **17.5** |

Table 8: Inception scores and FIDs with different reparametrization methods on CIFAR10 with the label supervision, by auxiliary classifier (Odena et al., 2017).

with respect to unnormalized weight $W$ as follows:

$$\frac{\partial V(G, D(W))}{\partial W} = \frac{1}{N(W)} \left( \frac{\partial V}{\partial \bar{W}} - \text{trace} \left( \left( \frac{\partial V}{\partial \bar{W}} \right)^{\text{T}} \bar{W} \right) \frac{\partial(N(W))}{\partial W} \right) \tag{33}$$

$$= \frac{1}{N(W)} \left( \nabla_{\bar{W}} V - \text{trace} \left( (\nabla_{\bar{W}} V)^{\text{T}} \bar{W} \right) \nabla_W N \right) \tag{34}$$

$$= \alpha \left( \nabla_{\bar{W}} V - \lambda \nabla_W N \right), \tag{35}$$

where $\alpha := 1/N(W)$ and $\lambda := \text{trace} \left( (\nabla_{\bar{W}} V)^{\text{T}} \bar{W} \right)$. The gradient $\nabla_{\bar{W}} V$ is calculated by $\hat{\text{E}} \left[ \boldsymbol{\delta} \boldsymbol{h}^{\text{T}} \right]$ where $\boldsymbol{\delta} := \left( \partial V(G, D) / \partial \left( \bar{W} \boldsymbol{h} \right) \right)^{\text{T}}$, $\boldsymbol{h}$ is the hidden node in the network to be transformed by $\bar{W}$ and $\hat{\text{E}}$ represents empirical expectation over the mini-batch. When $N(W) := \|W\|_F$, the derivative is:

$$\frac{\partial V(G, D(W))}{\partial W} = \frac{1}{\|W\|_F} \left( \hat{\text{E}} \left[ \boldsymbol{\delta} \boldsymbol{h}^{\text{T}} \right] - \text{trace} \left( \hat{\text{E}} \left[ \boldsymbol{\delta} \boldsymbol{h}^{\text{T}} \right]^{\text{T}} \bar{W} \right) \bar{W} \right), \tag{36}$$

and when $N(W) := \|W\|_2 = \sigma(W)$,

$$\frac{\partial V(G, D(W))}{\partial W} = \frac{1}{\sigma(W)} \left( \hat{\text{E}} \left[ \boldsymbol{\delta} \boldsymbol{h}^{\text{T}} \right] - \text{trace} \left( \hat{\text{E}} \left[ \boldsymbol{\delta} \boldsymbol{h}^{\text{T}} \right]^{\text{T}} \bar{W} \right) \boldsymbol{u}_1 \boldsymbol{v}_1^{\text{T}} \right). \tag{37}$$

Notice that, at least for the case $N(W) := \|W\|_F$ or $N(W) := \|W\|_2$, the point of this gradient is given by :

$$\nabla_{\bar{W}} V = k \nabla_W N. \tag{38}$$

where $^{\exists} k \in \mathbb{R}$

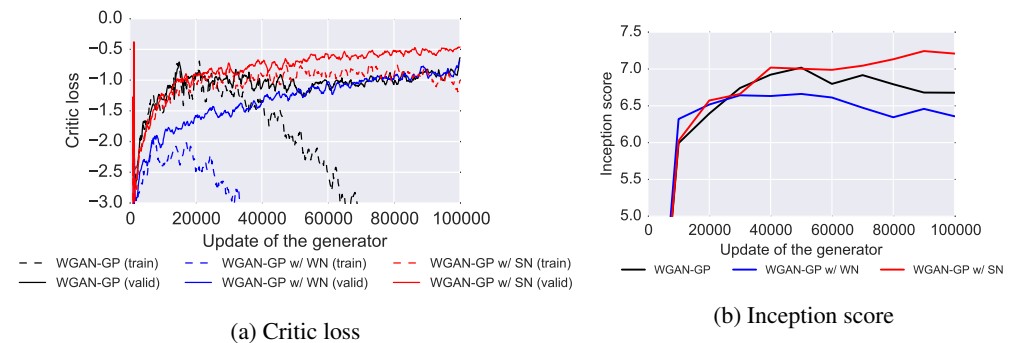

(a) Critic loss

(b) Inception score

Figure 15: Learning curves of (a) critic loss and (b) inception score on different reparametrization method on CIFAR-10 ; weight normalization (WGAN-GP w/ WN), spectral normalization (WGAN-GP w/ SN), and parametrization free (WGAN-GP).

