# OpenReview forum: "Spectral Normalization for Generative Adversarial Networks"
_ICLR.cc/2018/Conference — Accept (Oral)_

### Official Review · AnonReviewer2 · 2017-11-28
**Standard idea, great results**

**Rating:** 7
**Confidence:** 4

**Review:**

This paper borrows the classic idea of spectral regularization, recently applied to deep learning by Yoshida and Miyato (2017) and use it to normalize GAN objectives. The ensuing GAN, coined SN-GAN, essentially ensures the Lipschitz property of the discriminator. This Lipschitz property has already been proposed by recent methods and has showed some success. However,  the authors here argue that spectral normalization is more powerful; it allows for models of higher rank (more non-zero singular values) which implies a more powerful discriminator and eventually more accurate generator. This is demonstrated in comparison to weight normalization in Figure 4. The experimental results are very good and give strong support for the proposed normalization.


While the main idea is not new to machine learning (or deep learning), to the best of my knowledge it has not been applied on GANs. The paper is overall well written (though check Comment 3 below), it covers the related work well and it includes an insightful discussion about the importance of high rank models. I am recommending acceptance, though I anticipate to see a more rounded evaluation of the exact mechanism under which SN improves over the state of the art. More details in the comments below.

Comments:
1. One concern about this paper is that it doesn’t fully answer the reasons why this normalization works better. I found the discussion about rank to be very intuitive, however this intuition is not fully tested.  Figure 4 reports layer spectra for SN and WN. The authors claim that other methods, like (Arjovsky et al. 2017) also suffer from the same rank deficiency. I would like to see the same spectra included.
2. Continuing on the previous point: maybe there is another mechanism at play beyond just rank that give SN its apparent edge? One way to test the rank hypothesis and better explain this method is to run a couple of truncated-SN experiments. What happens if you run your SN but truncate its spectrum after every iteration in order to make it comparable to the rank of WN? Do you get comparable inception scores? Or does SN still win?
3. Section 4 needs some careful editing for language and grammar.

---

> ### Author Response · Authors · 2017-12-10
> **Response to AnonReviewer2**
>
>
> Thank you so much for the review!
>
> >This paper borrows the classic idea of spectral regularization, recently applied to deep learning by Yoshida and Miyato (2017) and use it to normalize GAN objectives.
>
> Thank you very much for the comments;  we however would like to remind that the spectral normalization presented in this paper is very much different from spectral ‘norm’ regularization introduced in Yoshida and Miyato (2017).
> Also,  unlike what we refer to as  “gradient penalty method”,  we are not regularizing the objective function in any means, so “normalize GAN objectives“ is an inaccurate keyword for our paper.
> We are still using the same objective function as the classic GAN; we are just looking for the candidate discriminator from the normalized set of functions.
> We will emphasize these points in the revised manuscript, since these confusions seem to be recurring issues.
>
>
> >1.  The authors claim that other methods, like (Arjovsky et al. 2017) also suffer from the same rank deficiency. I would like to see the same spectra included.
>
> Thanks for the suggestion. We plan to test with the weight clipping method  (Arjovsky et al. 2017) and report the results in the revised manuscript.
>
>
> >2. Continuing on the previous point: maybe there is another mechanism at play beyond just rank that give SN its apparent edge? One way to test the rank hypothesis and better explain this method is to run a couple of truncated-SN experiments. What happens if you run your SN but truncate its spectrum after every iteration in order to make it comparable to the rank of WN? Do you get comparable inception scores? Or does SN still win?
>
> That sounds like a good suggestion. Should there be ample rooms in the computational resource and time, we might try the experiment with CIFAR 10.
>
>
> >3. Section 4 needs some careful editing for language and grammar.
> Thanks, we will proofread the document once again.

---

### Official Review · AnonReviewer1 · 2017-11-30
**Nice step forward in improving training of GANs**

**Rating:** 8
**Confidence:** 3

**Review:**

This paper proposes "spectral normalization" -- constraining the spectral norm of the weights of each layer -- as a way to stabilize GAN training by in effect bounding the Lipschitz constant of the discriminator function. The paper derives efficient approximations for the spectral norm, as well as an analysis of its gradient. Experimental results on CIFAR-10 and STL-10 show improved Inception scores and FID scores using this method compared to other baselines and other weight normalization methods.

Overall, this is a well-written paper that tackles an important open problem in training GANs using a well-motivated and relatively simple approach. The experimental results seem solid and seem to support the authors' claims. I agree with the anonymous reviewer that connections (and differences) to related work should be made clearer. Like the anonymous commenter, I also initially thought that the proposed "spectral normalization " is basically the same as "spectral norm regularization", but given the authors' feedback on this I think the differences should be made more explicit in the paper.

Overall this seems to represent a strong step forward in improving the training of GANs, and I strongly recommend this paper for publication.

Small Nits:

Section 4: "In order to evaluate the efficacy of our experiment": I think you mean "approach".

There are a few colloquial English usages which made me smile, e.g.
 * Sec 4.1.1. "As we prophesied ...", and in the paragraph below
 * "... is a tad slower ...".

---

> ### Author Response · Authors · 2017-12-10
> **Response to AnonReviewer 1**
>
>
> Thank you so much for the review!
>
> >I also initially thought that the proposed "spectral normalization " is basically the same as "spectral norm regularization", but given the authors' feedback on this I think the differences should be made more explicit in the paper.
>
> Thanks for the suggestion; we will emphasize the difference between spectral norm regularization and our spectral normalization in the revised manuscript.
>
>
> And thanks for pointing out the colloquialism, we will relax it :-)

---

### Official Review · AnonReviewer3 · 2017-12-01
**Paper review: the methodology is neat, but presentation can be improved**

**Rating:** 7
**Confidence:** 2

**Review:**

The paper is motivated by the fact that in GAN training, it is beneficial to constrain the Lipschitz continuity of the discriminator. The authors observe that the product of spectral norm of gradients per each layer serves as a good approximation of the overall Lipschitz continuity of the entire discriminating network, and propose gradient based methods to optimize a "spectrally normalized" objective.

I think the methodology presented in this paper is neat and the experimental results are encouraging. However, I do have some comments on the presentation of the paper:

1. Using power method to approximate matrix largest singular value is a very old idea, and I think the authors should cite some more classical references in addition to (Yoshida and Miyato). For example,

Matrix Analysis, book by Bhatia
Matrix computation, book by Golub and Van Loan.

Some recent work in theory of (noisy) power method might also be helpful and should be cited, for example,
https://arxiv.org/abs/1311.2495

2. I think the matrix spectral norm is not really differentiable; hence the gradients the authors calculate in the paper should really be subgradients. Please clarify this.

3. It should be noted that even with the product of gradient norm, the resulting normalizer is still only an upper bound on the actual Lipschitz constant of the discriminator. Can the authors give some empirical evidence showing that this approximation is much better than previous approximations, such as L2 norms of gradient rows which appear to be much easier to optimize?

---

> ### Author Response · Authors · 2017-12-10
> **Response to AnonReviewer3**
>
>
> Thank you so much for the review!
>
>
> >The authors observe that the product of spectral norm of gradients per each layer serves as a good approximation of the overall Lipschitz continuity of the entire discriminating network, and propose gradient based methods to optimize a "spectrally normalized" objective.
>
> Thank you very much for the comments;  however we would like to emphasize that we are controlling the spectral norm of the operators, not their gradient.  Also, unlike what we refer to as  “gradient penalty method”,  we are not modifying the objective function in any means. We are still using the same objective function as the classic GAN; we are just looking for the candidate discriminator from the normalized set of functions.
>
>
> > 1. I think the authors should cite some more classical references in addition to (Yoshida and Miyato).
>
> Thanks for the remark, and yes we should have cited some of the classic references. We will add them to the revised manuscripts.
>
>
> >2. I think the matrix spectral norm is not really differentiable; hence the gradients the authors calculate in the paper should really be subgradients. Please clarify this.
>
> Indeed,  when the spectrum has multiplicities, we would be looking at subgradients, and technically we should have said so. However, the probability of this happening is zero (almost surely), and we assumed we can continue discussions without giving considerations to such events.  We will make note of this fact in the revised version.  Thanks!
>
>
> >3.  Can the authors give some empirical evidence showing that this approximation is much better than previous approximations, such as L2 norms of gradient rows which appear to be much easier to optimize?
>
> We would like to remind that the gest of our paper is not about the accuracy of the Lipschitz constant;  we do not intend to claim that our spectral normalization better controls the Lipschitz constant than the gradient penalty method.
> As we claim in Section 3, an advantage of our normalization over the gradient penalty based method (WGAN-GP) is that we can control the Lipschitz constant even outside the neighborhoods of the observed datapoint.
> Furthermore, spectral normalization can be carried out with less computational cost. Please see the discussions in the designated section for more detail.

---

### Public Comment · ~Ian_Goodfellow1 · 2017-11-20
**This is a great paper!**

This is a great paper! I don't think this paper explains the importance of its results nearly enough and I'm concerned that it may not be obvious what a breakthrough it is just from skimming the abstract.

"We tested the efficacy of spectral normalization on CIFAR10, STL-10, and ILSVRC2012 dataset, and we experimentally confirmed that spectrally normalized GANs (SN-GANs) is capable of generating images of better or equal quality relative to the previous training stabilization techniques" is a major understatement. This paper represents an extraordinary advance on the ILSVRC2012 dataset.

Before this paper, there was only one GAN that worked very well at all on ILSVRC2012: AC-GAN. AC-GAN was sort of cheating because it divided ImageNet into 100 smaller datasets that each contained only 10 classes. The new SN-GAN is the first GAN to ever fit all 1000 ImageNet classes in one GAN.

Scaling GANs to a high amount of classes has been a major open challenge and this paper has achieved an amazing 10X leap forward.

---

> ### Public Comment · (anonymous) · 2017-11-21
> **The reason why ILSVRC2012 is fundamentally hard**
>
> This may be a naive question, but can someone explain to me why scaling to ILSVRC2012 dataset is more than a computation problem? Is it because of the instability so that few realistic images will be generated or training progresses real slow? Or is it because mode collapsing so that less diverse set of realistic images will be generated? Or something else?

---

> > ### Public Comment · ~Ian_Goodfellow1 · 2017-11-21
> > **it's mode collapse**
> >
> > The main *symptom* is mode collapse.
> >
> > When I was working on this paper ( https://arxiv.org/abs/1606.03498 ) I was able to get GANs to draw dogs occasionally. Sometimes I could get them to draw a different class but I never got them to draw more than one class at a time.
> >
> > Augustus did some good experiments while working on AC-GAN, where he uses MS-SSIM to measure mode collapse. He found that increasing the number of classes causes collapse. https://arxiv.org/abs/1610.09585
> >
> > Of course, these are *symptoms*. We don't know a lot about the *cause*. There has been a lot of work over the past few years suggesting that the cause is optimizing the wrong loss (f-GAN, WGAN, both kinds of LS-GAN, etc. have proposed new losses). There has also been a lot of work suggesting the cause is that the learning algorithm fails to equilibrate the game or does so extremely inefficiently ( https://arxiv.org/abs/1412.6515 https://arxiv.org/abs/1701.00160 https://arxiv.org/abs/1706.04156 https://arxiv.org/abs/1706.08500 etc). Finally, at the MILA summer school this year, I said that I think the model family could bias the learning algorithm toward mode collapse. The success of SN-GAN in this submission seems to be evidence in favor of the 2nd or 3rd hypothesis about the cause.

---

> > > ### Public Comment · ~R_Devon_Hjelm1 · 2017-11-27
> > > **ILSVRC2012 collapse**
> > >
> > > Have there been any solid attempts at training Imagenet with WGAP-GP? This was not done in Gulrajani et al 2017, nor have I seen many recent attempts in the literature, even in the conditional setting. In my own experience I've found that the gradient norm regularization (https://arxiv.org/abs/1705.09367) seems to train reasonably well (no mode collapse, good diversity, some semblance of realistic samples, good quality as far as the usual features) with ResNets that usually fail with GANs, though I have to admit I'm in the middle of such an experiment and the samples still look quite strange at 20 epochs (like Dali paintings, this is taking weeks to train on one GPU).

---

> > > > ### Public Comment · ~Ian_Goodfellow1 · 2017-11-29
> > > > **I don't know of one**
> > > >
> > > > Because people don't usually release negative results, it's hard to know whether people have tried WGAN-GP on high-res ImageNet and it didn't work or whether no one has seriously tried it.
> > > >
> > > > I tried WGAN with weight clipping, not GP, on 128x128 ImageNet, by modifying the openai/improved-gan implementation of Minibatch (NS)GAN for the same dataset. WGAN with weight clipping didn't work very well for me on that task.
> > > >
> > > > This recent work suggests that WGAN-GP would probably perform comparably to NS-GAN: https://arxiv.org/abs/1711.10337

---

> > > > > ### Public Comment · ~Behrooz_Shahsavari1 · 2017-12-07
> > > > > **How about infogan?**
> > > > >
> > > > > It seems to me that a method like infogan that enforces a multi-modal distribution on the latent variables and makes the generator produce samples with high mutual-information with them can potentially solve the mode collapse issue. I am not sure how practical it is to train infogan for 1000 classes though. Do you see any reason that it cannot maintain the same number of image modes (classes) as the latent variable modes?

---

### Public Comment · ~Ian_Goodfellow1 · 2017-11-21
**Clarifications: implementing the power method**

I want to check that I understand how to implement the convolutional version of your spectral norm approximation correctly.

I make u be a convolutional input tensor that contains only one example:

single_input_shape = [1, rows, cols, input_channels]
self.single_input_shape = single_input_shape
init_u = tf.random_normal(single_input_shape, dtype=tf.float32)
init_u = init_u / tf.sqrt(1e-7 + tf.reduce_sum(tf.square(init_u)))
self.u = tf.Variable(init_u, trainable=False)


Then on every iteration of SGD I do these updates:

    new_v = self.conv(self.u, self.kernels)
    new_v = new_v / tf.sqrt(1e-7 + tf.reduce_sum(tf.square(new_v)))

    new_u = self.conv_t(new_v, self.kernels)
    # u^T W v = (W v / l2_norm(Wv))^T Wv = l2_norm(Wv) = l2_norm(new_u)
    spectral_norm = tf.sqrt(1e-7 + tf.reduce_sum(tf.square(new_u)))
    new_u = new_u / spectral_norm

   power_method_update =  tf.assign(self.u, new_u)


I ask because there are a few subtle things:
- I'm not sure which time step I'm meant to take u and v from when computing the spectral norm. Here I chose to use the *new* value of both u and v, so that get u^T W v for free when I compute the normalizing constant for the new value of u.
- For convolution, I *think* I'm meant to use convolution and convolution transpose on a 4-D tensor, based on the comment in the paper about the sparse matrix, but I wasn't totally sure if I should do this or reshape the kernels into a matrix and use matrix-vector products.
- I'm not 100% sure when I'm meant to run the `power_method_update` op. Should I just run this once per gradient step or do I need to run it several times to get u close to optimal before I start running SGD?

Thanks, and sorry if this is in the paper and I've missed it.

---

> ### Author Response · Authors · 2017-11-22
> **Thanks for the comments and remarks!**
>
> Thanks for the comments and remarks!
> Let me try to resolve the concerns one by one.
>
> >I'm not sure which time step I'm meant to take u and v from when computing the spectral norm. Here I chose to use the *new* value of both u and v, so that get u^T W v for free when I compute the normalizing constant for the new value of u.
>
> I am not sure if I am understanding the question clearly, but at each forward propagation, we prepare new u and v from the same set of u.
> By the way, we would like to note that we didn't propagate gradients thorough new_u and new_v .
> If we write our code in Tensorflow, our implementation is like:
>        new_v = tf.nn.l2_normalize(tf.matmul(self.u, W), 1)
>        new_u = tf.nn.l2_normalize(tf.matmul(new_v, tf.transpose(W)), 1)
>        new_u = tf.stop_gradient(new_u)
>        new_v = tf.stop_gradient(new_v)
>        spectral_norm = tf.reduce_sum(new_u * tf.transpose(tf.matmul(W, tf.transpos e(new_v))), 1)
>        power_method_update =  tf.assign(self.u, new_u)
>
> >For convolution, I *think* I'm meant to use convolution and convolution transpose on a 4-D tensor, based on the comment in the paper about the sparse matrix, but I wasn't totally sure if I should do this or reshape the kernels into a matrix and use matrix-vector products.
>
> In our implementations, we reshaped the 4D convolutional kernel into a 2-D matrix for the computation of the spectral norm. So, to be honest, our “spectral norm” does not include the parameters like padding and stride size.  We did away with these parameters just for the ease of computation.   So far however, this way is yielding satisfactory results.
>
> Your implementation is mathematically more faithful to our theoretical statement in that it is approximating the honest-to-goodness operator norm of the convolutional operator that includes these parameters.  We cannot say for 100% sure, but your speculate your way of computation shall work just fine.
>
> >I'm not 100% sure when I'm meant to run the `power_method_update` op. Should I just run this once per gradient step or do I need to run it several times to get u close to optimal before I start running SGD?
>
> In our experiment, we applied the power method update operation only one time per gradient step. It turned out that one power iteration was enough.
> To check how good we are doing with one application of the power method,  we used SVD to compute the spectral norm of the convolution kernel normalized with our method (AppendixC.1)  Note that our method is doing just fine with  one power method.

---

> > ### Public Comment · ~Ian_Goodfellow1 · 2017-11-22
> > **Thanks!**
> >
> > Thanks! Yes, I see from Appendix C1 that you're finding a good approximation of the spectral norm. I was asking these questions so I can re-implement it successfully myself, not because I doubt you're finding the spectral norm.
> >
> > Two follow up questions:
> > 1)
> > If the spectral norm
> > sigma(W) = u^T W v
> > then to estimate the derivatives of sigma(W) with respect to W, don't you need to backprop through u and v too? u and v are both functions of W.
> > Or is the estimate still useful somehow when u and v are constant?
> > (Either way, you successfully maintain the spectral norm constraint, but learning would be faster if you get the gradient of sigma(W) correct because this means your gradient of the cost function will be tangent to the constraint region and prevent you from wasting time moving in forbidden directions)
> >
> > 2)
> > For convolution, you have a kernel K that can be reshaped to a matrix W but convolving the input actually uses a different matrix B. B is a big doubly block circulant matrix where the number of rows is equal to the number of pixels in the input image.
> >
> > sigma(B) is max_{image, subject to l2_norm(image)=1} l2_norm(conv(image, K)).
> >
> > sigma(W) is max_{x, subject to l2_norm(x)=1} l2_norm(Wx).
> >
> > Do you know if sigma(B) and sigma(W) are approximately the same  as each other?
> > I haven't thought it through and don't actually know the answer.
> >
> > To bound the Lipschitz constant of the neural net, you want to constrain sigma(B), but in your experiments you constrained sigma(W).
> >
> > I'm guessing that maybe sigma(B) and sigma(W) are related by a constant factor, so you probably are still constraining the Lipschitz constant of the whole net, but maybe constraining it to a different value than you thought.

---

> > > ### Author Response · Authors · 2017-11-29
> > > **backprop, the spectral norm of the convoluation operation**
> > >
> > > 1)
> > > Indeed, u and v are both functions of W, and we technically have to backprop through these vectors as well.  However,  in our implementation, we ignored the dependency of u and v on W for the sake of computational efficiency, and we were still able to maintain the Lipschitz constraint.
> > > In fact, to be on the safe side, we ran experiments with backprop on u and v as a separate experiment.  We were not able to observe any notable improvement.
> > >
> > > 2)
> > > To make the long story short,  sigma(W) and sigma(B) may and may not differ depending on the padding and stride size.  We briefly discuss this matter on the second footnote in page 5.  Let us elaborate on this a little further.  For the sake of argument, let us assume that the input image is infinite dimensional in both directions. If the stride size is 1,  the value on each output pixel will be computed from the outputs of exactly same number (say, m) of filter blocks.  The same holds also when the stride size divides the dimension of the filter block. In such cases, sigma(W) and sigma(B) will be off by the root m, and the dominant vectors will be exactly same.
> > > When the stride size does not divide the dimension of the dimension of the filter block, however, there will be some output pixels that are computed from the outputs of more filter blocks than other. In such cases, the relationship between sigma(W) and sigma(B) appears complex; at least so complex that we decided not to elaborate further on our paper.
> > > For our experiment, we made sure that the stride size divides the dimension of the filter block so that, even after taking the padding size into consideration, the dominant direction will not be too much off from what we mathematically intended.

---

> > > > ### Public Comment · ~Rémi_LE_PRIOL1 · 2018-05-10
> > > > **This is projected gradient**
> > > >
> > > > Hello,
> > > >
> > > > I really enjoyed your paper. Yet I observed some discrepancy between your code and the maths in your paper.
> > > >
> > > > In your code, you don't back-propagate through the spectral norm. You simply compute it, then divide the matrix by it. It means that you are simply 'projecting' the matrix on the l^2 unit sphere of the matrix space. You could do this at the optimization time really.
> > > >
> > > > Yet the way you talk about the gradient of the normalization layer in your paper led me to think that you are back-propagating through the normalization. You could do this without back-propagating through the whole power iteration by specifying a backward operator for the layer I guess.
> > > >
> > > > So what is the truth?

---

> > > > > ### Author Response · Authors · 2018-05-21
> > > > > **Re: This is projected gradient**
> > > > >
> > > > > Hi,  thanks for your interest in our work!
> > > > >
> > > > > We actually propagate the gradient through the spectral norm.
> > > > >
> > > > > The line :https://github.com/pfnet-research/sngan_projection/blob/master/source/functions/max_sv.py#L30
> > > > > calculates the spectral norm given W.
> > > > > When we apply the backprop, the gradient will propagate to W, which is the weight parameter of linear / conv layer.

---

> > > > ### Public Comment · (anonymous) · 2018-12-22
> > > > **sigma(W) vs sigma(B)**
> > > >
> > > > Are you saying that in some very good special case (i.e. when stride=1 and appropriate padding is used) we really have $\sigma(W)=\sigma(B)$? This apparently wouldn't be true.
> > > >
> > > > The fact that one is bounded by a constant factor of another might be a trivial consequence of finite-dimensionality, as both $W\mapsto \sigma(\operatorname{reshape}(W))$ and $W\mapsto \sigma(B)$ seem to define valid norms on kernels

---

### Public Comment · ~Colin_Raffel1 · 2017-11-22
**Possible typo when describing convolution filterbank flattening?**

Hi, thanks for the paper, impressive results!

I am confused about how you describe flattening the convolutional filterbank for computing the spectral norm.  You wrote
"Also, for the evaluation of the spectral norm for the convolutional weight $W \in R^{d_{out} \times d_{in} \times h \times w}$, we treated the operator as a square matrix of dimension $d_{out} \times (d_{in}hw)^2$."
I don't see where the square comes in.  If you flatten $W$, it should be of shape $d_{out}d_{in}hw$, right?  I am also interested to hear more about the semantics of the spectral norm of this object (flattened filterbank), which Ian asked about below.

Also, a separate question - for your imagenet experiments, did all of the GAN variants you report (no normalization, layer normalization, spectral normalization) have conditional batch norm?  Or just the SN-GAN?  Relatedly, I think there is a typo in the caption of Table 6:
"we replaced the usual batch normalization layer in the ResBlock of the with the conditional batch normalization layer"

Thanks for any responses.

---

> ### Author Response · Authors · 2017-11-29
> **Thanks for the comments!**
>
>
> >I don't see where the square comes in.  If you flatten $W$, it should be of shape $d_{out}d_{in}hw$, right?  I am also interested to hear more about the semantics of the spectral norm of this object (flattened filterbank), which Ian asked about below.
>
> Yes, it's a typo. We meant to write 2-D, not square.
> As for the spectral norm of convolutional operator, please take a look at our response to Ian’s comment.
>
> >Relatedly, I think there is a typo in the caption of Table 6:
> "we replaced the usual batch normalization layer in the ResBlock of the with the conditional batch normalization layer"
>
> We are sorry for the confusion, and you are correct about our typo in the caption of Table 6. We meant to write
> “we replaced the usual batch normalization layer in the ResBlock of the '''generator''' with the conditional batch normalization layer".
> We introduced the conditional batch normalization layer to the generators of ALL the GANs.

---

### Public Comment · ~Leon_Boellmann1 · 2017-12-06
**The Lipschitz constraint does not avoid mode collapse?**

 Dear authors,
 Your paper attracts my attention, because it has impressive results.
 May I know what is the motivation for Lipschitz constraint for GAN? For WGAN, the Lipschitz constraint is required due to the Wasserstein divergence metric. However, for the original GAN, I am not clear why the Lipschitz constraint is so helpful. According to my knowledge, one source of mode collapse is that the generated probability at some data points is very small (almost zero) and D(x) is locally constant (equal to 0) around those data points. Then the gradient of D(x) at those data points is zero, which prevents the update of the generator. Detailed description can be found in "Towards Principled Methods for Training Generative Adversarial Networks". In this case, even if D(x) satisfies the Lipschitz norm constraint, it still cannot avoid mode collapse. Would the author please clarify how spectral normalization helps avoid mode collapse?

---

> ### Author Response · Authors · 2017-12-20
> **Thanks for the comment!**
>
> We need to remember that the paper you designated uses the classic loss function for both “generator” and discriminator updates.  As we explain in the experiment section, we are using the modified generator updated rule proposed by Goodfellow et al (2014), which uses softplus function -log sigmoid(f(x)) = log  (1 + exp(-f(x))) := softplus(-f(x)) in place of log (1- sigmoid(f(x))) so that one can maintain the learning process.  Note that softplus(-f(x)) is approximately -f(x)  when f(x) < 0 (In fact, on the bulk of the support of the generator,  f(x) tends to be negative. ). Thus, the generator will be looking at the gradient of f(x) on the course of its training.  As such, we need to keep our eyes on the gradient of f(x), which can blow up outside of the support of p or q (see Eq (4) ) without any gradient regularization as a countermeasure.
> So far, this is our current postulate on the importance of Lipschitz constant in GAN. The gest of our paper is that, WGAN-GP and our spectral normalization can constrain the norm of the gradient so that this will not be a problem.
>
> Also, we shall make it clear that our method is not designed specifically for the purpose of preventing mode-collapse.  However, it is not hard to imagine that the control of the Lipschitz constant of the discriminator would prevent the training process of the generator to plateau prematurely because of the critical gradient problem we have described above.

---

> > ### Public Comment · ~Leon_Boellmann1 · 2017-12-20
> > **Thanks for the explanation.**
> >
> > Thanks for the explanation for the intuition of the spectral normailzation and gradient norm. It is clearer to me now.

---

### Public Comment · (anonymous) · 2017-12-25
**Please can you share the code to reproduce ILSVRC2012 imagenet results ?**

Please can you share the code to reproduce ILSVRC2012 imagenet results ?

---

> ### Author Response · Authors · 2017-12-26
> **Thanks for the comment.**
>
> Hi, thanks for your comment.
>
> We will share the reproducing code after the acceptance notification.
> We will announce the link to the code here when we make the code public.

---

### Author Response · Authors · 2017-12-27
**Uploaded the revision.**

We owe great thanks to all reviewers for helpful comments toward improving our manuscripts.
We revised our manuscript based on the reviewer’s comments (including the ones that were visible to us by mistake because of the administrator’s technical problem) and uploaded the revision.

Firstly,  we conducted additional comparative study against orthonormal regularization, and showed the advantage of our algorithm over the orthonormal regularization.

Secondly,  we responded to the AnonReviewer2’s comment by running still another experiment with weight clipping (Arjovsky et al. 2017) and compared the results on CIFAR10 and STL10.
We confirmed that, as we have noted in Section 3, weight clipping also suffered from the rank degeneracy and its performance turned out to be much worse than our spectral normalization.

Thirdly, for the ImageNet, we re-calculated the inception scores for all methods using the original tensorflow implementation and replaced the scores on the table, because we were using Chainer instead of Tensorflow exclusively for the ImageNet results, and there were some numerical variations. The newly computed values do not affect any of our claims regarding the advantages and the superiority of our algorithm.

---

### Public Comment · (anonymous) · 2018-01-04
**Variance Shift In Deep Network With Spectral Normalization**

HI, nice work!

Spectral normalization seems to be useful as a regularization for GAN training.

But I'm considering that spectral normalization cannot ensure "unshift of mean & variance" of the output of each layer. Usually, we would like the mean = 0 and variance = 1 for a deep network to avoid vanishing/explosion of activation and gradient.

Is this an issue with spectral normalization? and how should we handle it?

Thanks.

---

### Author Response · Authors · 2018-02-02
**The code for reproducing the results**

The code for reproducing the results in this paper has been uploaded at
https://github.com/pfnet-research/sngan_projection.
Also we have uploaded other materials (pretrainied models, generated images and movies) at https://drive.google.com/drive/folders/1GnDuF02F3a_zNEwiA74DnaG7OQ3-Co3N.
Please go to the links if you are interested in our work.

---

### Decision · Program_Chairs · 2018-01-29
**ICLR 2018 Conference Acceptance Decision**

**Decision:**

Accept (Oral)

**Comment:**

This paper presents impressive results on scaling GANs to ILSVRC2012 dataset containing a large number of classes. To achieve this, the authors propose "spectral normalization" to normalize weights and stabilize training which turns out to help in overcoming mode collapse issues.  The presented methodology is principled and well written. The authors did a good job in addressing reviewer's comments and added more comparative results on related approaches to demonstrate the superiority of the proposed methodology. The reviewers agree that this is a great step towards improving the training of GANs.  I recommend acceptance.